# CoDA: Collaborative Novel Box Discovery and Cross-modal Alignment for Open-vocabulary 3D Object Detection

**Yang Cao**[1]    **Yihan Zeng**[2]    **Hang Xu**[2]    **Dan Xu**[1*]
Hong Kong University of Science and Technology[1]
Huawei Noah's Ark Lab[2]

## Abstract

Open-vocabulary 3D Object Detection (OV-3DDet) aims to detect objects from an arbitrary list of categories within a 3D scene, which remains seldom explored in the literature. There are primarily two fundamental problems in OV-3DDet, *i.e.*, localizing and classifying novel objects. This paper aims at addressing the two problems simultaneously via a unified framework, under the condition of limited base categories. To localize novel 3D objects, we propose an effective 3D Novel Object Discovery strategy, which utilizes both the 3D box geometry priors and 2D semantic open-vocabulary priors to generate pseudo box labels of the novel objects. To classify novel object boxes, we further develop a cross-modal alignment module based on discovered novel boxes, to align feature spaces between 3D point cloud and image/text modalities. Specifically, the alignment process contains a class-agnostic and a class-discriminative alignment, incorporating not only the base objects with annotations but also the increasingly discovered novel objects, resulting in an iteratively enhanced alignment. The novel box discovery and cross-modal alignment are jointly learned to collaboratively benefit each other. The novel object discovery can directly impact the cross-modal alignment, while a better feature alignment can, in turn, boost the localization capability, leading to a unified OV-3DDet framework, named **CoDA**, for simultaneous novel object localization and classification. Extensive experiments on two challenging datasets (*i.e.*, SUN-RGBD and ScanNet) demonstrate the effectiveness of our method and also show a significant mAP improvement upon the best-performing alternative method by 80%. Codes and pre-trained models are released on the project page[1].

## 1   Introduction

3D object detection (3D-Det) [19, 21, 20] is a fundamental task in computer vision, with wide applications in self-driving, industrial manufacturing, and robotics. Given point cloud scenes, 3D-Det aims to localize and classify objects in the 3D scenes. However, most existing works heavily rely on datasets with fixed and known categories, which greatly limits their potential applications in the real world due to the various and unbounded categories. Open-vocabulary 3D object detection (OV-3DDet) is on the verge of appearing to detect novel objects. In OV-3DDet, the model is typically trained on datasets with limited base categories while evaluated on scenes full of novel objects, which brings additional challenges. In the problem of OV-3DDet, the key challenge is how to localize and classify novel objects using only annotations of very limited base categories. There are only very few works in the literature targeting this challenging problem so far.

---

[*]Corresponding author
[1] `https://yangcaoai.github.io/publications/CoDA.html`

37th Conference on Neural Information Processing Systems (NeurIPS 2023).

To obtain novel object boxes, [15] utilizes a large-scale pre-trained 2D open-vocabulary object detection (OV-2DDet) model [35] as prior knowledge to localize a large number of 2D novel object boxes, and then generates pseudo 3D object box labels for corresponding 3D novel objects. Instead of directly relying on an external OV-2DDet model to obtain novel object localization capability, we aim to learn a discovery of novel 3D object boxes, based on limited 3D annotations of base categories on the target data. With the discovered novel object boxes, we further explore joint class-agnostic and class-specific cross-modal alignment, using the human-annotated base object boxes, and the discovered novel object boxes. The novel object box discovery and cross-modal alignment are collaboratively learned to achieve simultaneous novel object localization and classification.

Specifically, to localize novel objects, we first propose a 3D Novel Object Discovery (3D-NOD) training strategy that utilizes information from both the 3D and 2D domains to discover more novel objects in the training process. In detail, the model learns class-agnostic 3D box prediction using 3D box annotations of the base categories, which can provide 3D geometry priors of 3D boxes based on geometric point cloud features. To confirm discovered novel boxes, the pretrained vision-language model CLIP [22] is also utilized to provide 2D semantic category priors. By considering both 3D and 2D cues, novel objects can be effectively discovered during iterative training.

Building upon 3D-NOD, to achieve novel object classification, we further propose a Discovery-Driven Cross-Modal Alignment (DCMA) module that aligns 3D object features with 2D object and text features in a large vocabulary space, using both human-annotated 3D object boxes of base categories, and the discovered 3D boxes of novel categories. DCMA comprises two major components, *i.e.*, category-agnostic box-wise feature distillation and class-specific feature alignment. The former aligns 3D point cloud features and 2D CLIP image features at the box level, without relying on information about specific categories. The distillation pushes the 3D point cloud and 2D image features closer wherever a 3D box covers. On the other hand, DCMA aligns 3D object features with CLIP text features in a large category vocabulary space, using category information from both annotated base objects and discovered novel objects, in a cross-modal contrastive learning manner.

The proposed 3D-NOD and DCMA can be jointly optimized in a unified deep framework named CoDA. They collaboratively learn to benefit each other. The discovered novel object boxes through 3D-NOD can greatly facilitate the cross-modal feature alignment, while better feature alignment through DCMA can further boost the model localization capability on novel objects. With our joint training of 3D-NOD and DCMA in CoDA to simultaneously localize and classify novel objects, our approach outperforms the best-performing alternative method by 80% in terms of mAP.

In summary, the main contribution of the paper is three-fold:

- We propose an end-to-end open vocabulary 3D object detection framework named **CoDA**, which can learn to simultaneously localize and classify novel objects, without requiring any prior information from external 2D open-vocabulary detection models.

- We design an effective 3D Novel Object Discovery (**3D-NOD**) strategy that can localize novel objects by jointly utilizing 3D geometry and 2D semantic open-vocabulary priors. Based on discovered novel objects, we further introduce discovery-driven cross-modal alignment (**DCMA**) to effectively align 3D point cloud features with 2D image/text features in a large vocabulary space.

- We design a mechanism to collaboratively learn the 3D-NOD and DCMA to benefit each other. The novel object boxes discovered from 3D-NOD are used in cross-modal feature alignment in DCMA, and a better feature alignment in DCMA can facilitate novel object discovery.

## 2   Related Work

**3D Object Detection.** 3D object detection (3D-Det) [21, 25, 32, 16, 33, 26, 3] has achieved great success in recent years. For instance, VoteNet [21] introduces a point voting strategy to 3D object detection. It adopts PointNet [20] to process 3D points and then assigns these points to several groups, and object features are extracted from these point groups. Then the predictions are generated from these object features. Based on that, MLCVNet [25] designs new modules for point voting to capture multi-level contextual information. Recently, the emergence of transformer-based methods [2] has also driven the development of 3D-Det. For example, GroupFree [14] utilizes a transformer as the prediction head to avoid handcrafted grouping. 3DETR [19] proposes the first end-to-end transformer-based structure for 3D object detection. However, these methods mainly focus on the

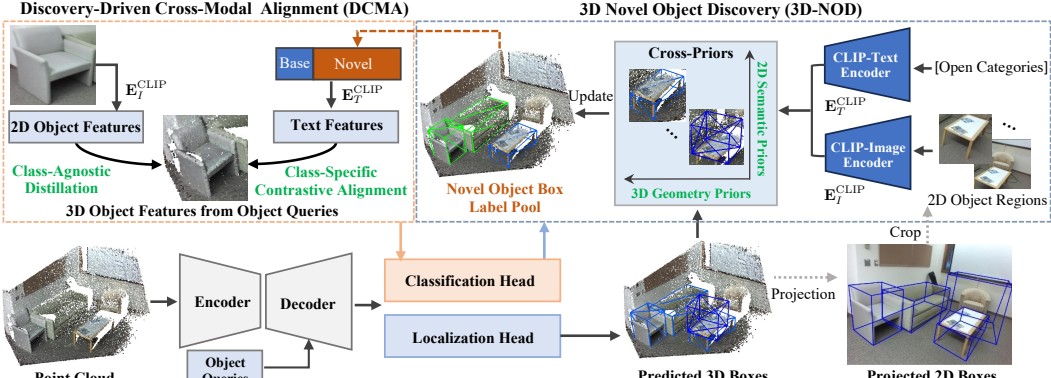

Figure 1: Overview of the proposed open-vocabulary 3D object detection framework named **CoDA**. We consider 3DETR [19] as our base 3D object detection framework, represented by the 'Encoder' and 'Decoder' networks. The object queries, together with encoded point cloud features, are input into the decoder. The updated object query features from the decoder are further input into 3D object classification and localization heads. We first propose a 3D Novel Object Discovery (**3D-NOD**) strategy, which utilizes both 3D geometry priors from predicted 3D boxes and 2D semantic priors from the CLIP model to discover novel objects during training. The discovered novel object boxes are maintained in a novel object box label pool, which is further utilized in our proposed discovery-driven cross-modal alignment (**DCMA**). The DCMA consists of a class-agnostic distillation and a class-specific contrastive alignment based on discovered novel boxes. Both 3D-NOD and DCMA collaboratively learn to benefit each other to achieve simultaneous novel object localization and classification in an end-to-end manner.

close-vocabulary setting, *i.e.*, the object categories in training and testing are the same and fixed. In contrast to these works, this paper targets open-vocabulary 3D-Det, and we design an end-to-end deep pipeline called CoDA, which performs joint novel object discovery and classification.

**Open-vocabulary 2D object detection.** In open-vocabulary 2D object detection (OV-2DDet), some works [7, 8, 10, 22, 11, 27, 18, 6, 17, 35, 29, 28] push 2D object detection to the open-vocabulary level. Specifically, RegionCLIP [34] pushes the CLIP from the image level to the regional level and learns region-text feature alignment. Detic [35] instead utilizes image-level supervision based on ImageNet [5], which assigns image labels to object proposals. GLIP [13] and MDETR [12] treat detection as the grounding task and adopt text query for the input image to predict corresponding boxes. All the methods mentioned above focus on 2D object detection and conduct the object detection from the 2D input image. However, in this work, we focus on open-world 3D object detection, which is a challenging problem and remains seldom explored in the literature.

**Open-vocabulary 3D object detection.** Recently, an open-set 3D-Det method [1] has shown great potential in various practical application scenarios. It can successfully detect unlabeled objects by introducing ImageNet1K [5] to train a 2D object detection model [2], which classifies objects generated by a 3D-Det model [19] on the training set to obtain pseudo labels for the objects. Then the pseudo labels are adopted to retrain the open-set 3D-Det model [19]. Even though there are no labels for the novel objects during training, the names of novel categories are known and fixed so that the category-specific pseudo labels can be generated, which is thus an open-set while close-vocabulary model. More recently, [15] propose a deep method to tackle OV-3Ddet. They adopt the CLIP model [22] and a large-scale pretrained external OV-2Ddet [35] model to generate pseudo labels of possible novel objects. Our work is closer to [15], but we target OV-3Ddet without relying on any additional OV-2Ddet model as priors to obtain novel box localization capabilities. Our model CoDA can simultaneously learn to conduct novel object discovery and cross-modal alignment with a collaborative learning manner, to perform end-to-end novel object localization and classification.

## 3 Methods

### 3.1 Framework Overview

An overview of our proposed open-world 3D object detection framework called **CoDA** is shown in Fig. 1. Our method utilizes the transformer-based point cloud 3D object detection model

3DETR [19] as the detection backbone, which is represented as an 'Encoder' and a 'Decoder' shown in Fig. 1. It has localization and a classification head for 3D object box regression and box category identification, respectively. To introduce open-vocabulary detection capabilities to the 3D point cloud object detection task, we utilize the pretrained vision-language model CLIP [22] to bring us rich open-world knowledge. The CLIP model consists of both an image encoder (*i.e.*, CLIP-Image) and a text encoder (*i.e.*, CLIP-Text) in Fig. 1. To localize novel objects, we first propose the 3D Novel Object Discovery (**3D-NOD**) strategy, which jointly utilizes both 3D box geometry priors from base categories and 2D semantic priors from the CLIP model to identify novel objects. To classify novel objects, we further propose a novel discovery-driven cross-modal alignment (**DCMA**) module, which aligns feature spaces between 3D point clouds and image/text modalities, using the discovered novel object boxes as guidance. The 3D-NOD and DCMA are jointly optimized to collaboratively learn to benefit each other in the unified deep framework.

## 3.2   3D Novel Object Discovery (3D-NOD) with Cross-Priors

As shown in Fig. 1, our 3D Novel Object Discovery (3D-NOD) strategy allows for the discovery of novel objects during training. To achieve this goal, we utilize cross-priors from both 3D and 2D domains. In terms of the 3D domain, we use 3D box geometry prior provided by the 3D object annotations of base categories, to train a class-agnostic box predictor. Regarding the 2D domain, we utilize the semantic priors from the CLIP model, to predict the probability of a 3D object belonging to a novel category. We combine these two perspectives to localize novel object boxes.

**Discovery based on 3D Geometry Priors.** We begin with an initial 3D object box label pool $\mathbf{O}_0^{base}$ for objects of base categories (*i.e.*, seen categories) in the training set:

$$\mathbf{O}_0^{base} = \left\{ o_j = (l_j, c_j) \mid c_j \in \mathbb{C}^{Seen} \right\}, \tag{1}$$

where $\mathbb{C}^{Seen}$ represents the base category set of the training data, which has human-annotated object labels. Based on $\mathbf{O}_0^{base}$, we can train an initial 3D object detection model $\mathbf{W}_0^{det}$ by minimizing an object box regression loss function derived from 3DETR [19]. We consider a class-agnostic object detection for $W_0$, meaning that we train the model with class-agnostic box regression and binary objectness prediction, and do not utilize the class-specific classification loss. This is because the class-specific training with strong supervision from base categories hampers the model's capabilities of discovering objects from novel categories, which has also been observed from previous OV-2DDet frameworks [8]. Thus our modified backbone outputs object boxes, including the parameters for the box 3D localization, and their objectness probabilities. For the $n$-th object query embedding in 3DETR, we predict its objectness probability $p_n^g$, based on $\mathbf{W}_0^{det}$ which is learned using 3D geometry priors from human-annotated 3D object boxes on base categories.

**Discovery based on 2D CLIP Semantic Priors.** Then, we also project the 3D object box $l_n^{3D}$ in the point cloud to the 2D box $l_n^{2D}$ on the color image with the corresponding camera intrinsic matrix $M$ as follows:

$$l_n^{2D} = M \times l_n^{3D}, \tag{2}$$

We then obtain the 2D object region $I_n^{2D}$ in the color image by cropping it with $l_n^{2D}$. Next, we obtain the 2D object feature $\mathbf{F}_{I,n}^{Obj}$ by passing the object region into the CLIP image encoder $\mathbf{E}_I^{CLIP}$. As the testing categories are unknown during training, we adopt a super category list $T^{super}$ following [9] to provide text semantic descriptions of rich object categories. We encode $T^{super}$ with the CLIP text encoder $\mathbf{E}_T^{CLIP}$ to obtain the text embedding $\mathbf{F}_T^{Super}$. We then estimate the semantic probability distribution (*i.e.*, $\mathbf{P}_n^{3DObj}$) of the 3D object using its corresponding 2D object feature as:

$$\mathbf{P}_n^{3dObj} = \left\{ p_{n,i}^s, p_{n,2}^s, p_{n,3}^s, ..., p_{n,C}^s \right\} = \text{Softmax}(\mathbf{F}_{I,n}^{Obj} \cdot \mathbf{F}_T^{Super}), \tag{3}$$

where $C$ is the number of categories defined in the super category set $T^{super}$. The symbol $\cdot$ indicates the dot product operation. $\mathbf{P}^{3dObj}$ provides semantic priors from the CLIP model. We can obtain the object category $c^*$ for the object query by finding the maximum probability in $\mathbf{P}^{3dObj}$ as $c^* = \text{argmax}_c \mathbf{P}^{3DObj}$. We then combine both the objectness $p_n^g$ from the 3D geometry priors and $p_{n,c^*}^s$ from the 2D semantic priors, and discover novel objects for the $t$-th training epoch as follows:

$$\mathbf{O}_t^{disc} = \left\{ o_j \mid \forall o_i' \in \mathbf{O}_0^{base}, \text{IoU}_{3D}(o_j, o_i') < 0.25, p_n^g > \theta^g, p_{n,c^*}^s > \theta^s, c^* \notin \mathbb{C}^{Seen} \right\}, \tag{4}$$

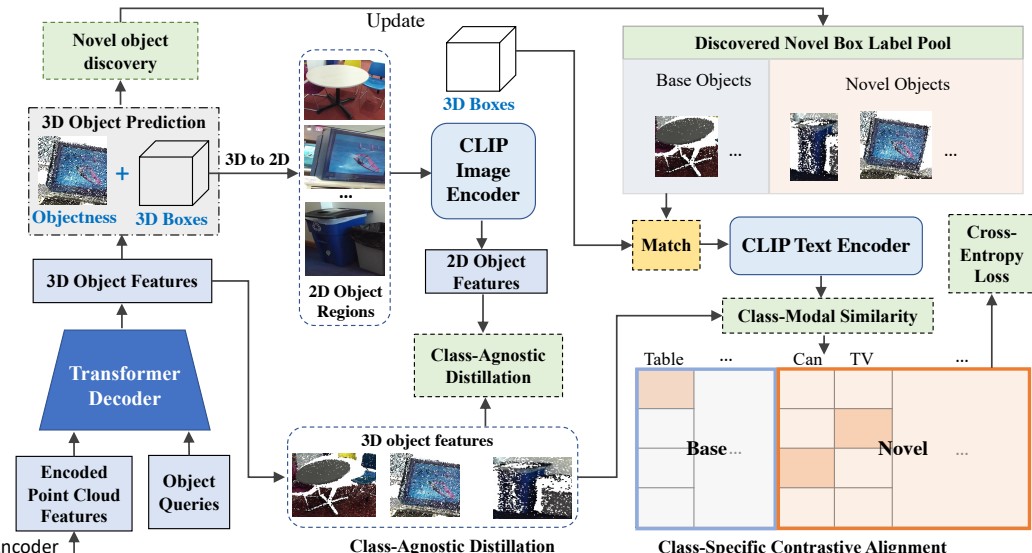

Figure 2: Illustration of the proposed discovery-driven cross-modal alignment (**DCMA**) in our OV detection framework (**CoDA**). The DCMA consists of two parts, *i.e.*, the class-agnostic distillation and the class-specific contrastive feature alignment. The predicted 3D boxes from the detection head are projected to obtain 2D image object regions, which are passed into the CLIP image encoder to generate 2D object features. Then, the CLIP 2D object features and 3D point cloud object features are fed into the class-agnostic distillation module for feature alignment. Driven by discovered novel object boxes from the 3D-NOD strategy, the maintained novel box label pool that is updated during training can be used to match with the predicted 3D object boxes. We perform contrastive alignment for matched novel boxes to learn more discriminative 3D object features for novel objects. More discriminative 3D object features can in turn facilitate the prediction of novel object boxes.

where $\text{IoU}_{3D}$ means the calculation of the 3D IoU of two boxes; $\theta^s$ and $\theta^g$ are two thresholds for the semantic and the geometry prior, respectively; $\mathbb{C}^{seen}$ is the set of seen categories of the training data. Then, the discovered novel objects $O_t^{disc}$ are added to a maintained 3D box label pool $\mathbf{O}_t^{novel}$ as:

$$\mathbf{O}_{t+1}^{novel} = \mathbf{O}_t^{novel} \cup \mathbf{O}_t^{disc}, \tag{5}$$

The model is trained to iteratively update the novel object box label pool. By gradually expanding the box label pool with the novel objects $O_t^{novel}$, we can broaden the model's capabilities to localize novel objects by using supervision from $O_t^{novel}$ for the model $W_t$. With the semantic priors from CLIP and the 3D geometry prior from $W_t$, the model and the 3D object box label pool can discover more and more novel objects following the proposed object discovery training strategy.

### 3.3 Discovery-Driven Cross-Modal Alignment (DCMA)

To bring open-vocabulary capabilities from the CLIP model, we propose a discovery-driven cross-modal alignment (DCMA) module, to align the 3D point cloud object features with 2D-image/CLIP-text features, based on the discovered novel object boxes $\mathbf{O}^{novel}$. For the proposed DCMA, we utilize the discovered novel object boxes to conduct two types of cross-modal feature alignment, *i.e.*, class-agnostic distillation and class-specific contrastive alignment.

**Object-Box Class-agnostic Distillation.** We introduce a class-agnostic distillation strategy to align the 3D point cloud object features and 2D image object features, to transfer more open-world knowledge from the CLIP model. For the $n$-th object query, we can obtain its updated 3D feature $\mathbf{F}_n^{3dObj}$ at the last layer of the decoder. Based on the 3D object feature, the model predicts its 3D box parameterization $l_n^{3D}$, and then we can obtain its corresponding 2D box $l_n^{2D}$ by Eqn. (2). Now we have 2D-projected object features $\mathbf{F}_n^{2DObj}$ by inputting CLIP with the cropped 2D region based on $l^{2D}$. To push the feature spaces of both the 3D and 2D objects as close as possible, if a 3D object box is predicted from the 3D object feature, we align them by using a class-agnostic L1 distillation loss,

which is defined as follows:

$$\mathcal{L}_{distill}^{3D\leftrightarrow2D} = \sum_{n=1}^{N} \|\mathbf{F}_n^{3DObj} - \mathbf{F}_n^{2DObj}\|, \tag{6}$$

where $N$ is the number of object queries. As shown in Fig. 2, even if the box covers a background region, the class-agnostic distillation still narrows the distance between the two modalities, leading to a more general alignment on different modalities and scenes. As we can see, class-agnostic distillation eliminates the requirement of category labels of GT boxes. Thus, the computation of $\mathcal{L}_{distill}^{3D\leftrightarrow2D}$ does not need category annotations.

**Discovery-driven class-specific contrastive alignment.** As we discussed in Sec. 3.2, we modify the 3DETR detection backbone, by removing the fixed-category class-specific classification head. We can extract 3D point cloud object features from the updated object query embeddings. For the $n$-th object query embedding, we can obtain a corresponding 3D object feature $\mathbf{F}_n^{3DObj}$ at the last layer of the decoder. With the 3D object feature $\mathbf{F}_n^{3DObj}$, we expect the feature to be discriminative for both base categories and novel categories defined in the superset $T^{super}$. We, therefore, perform class-specific alignment with the CLIP text features. Specifically, we calculate the normalized similarities using the dot product operation between $\mathbf{F}_n^{3DObj}$ and the text features $\mathbf{F}_T^{Super}$ as follows:

$$\mathbf{S}_n = \text{Softmax}(\mathbf{F}_n^{3DObj} \cdot \mathbf{F}_T^{Super}), \tag{7}$$

Then, we introduce the point cloud and text contrastive loss to provide the supervision signal for learning the 3D object features of the object queries. We have already measured the similarity between the 3D object point cloud feature and the text features. To construct a cross-modal contrastive loss, we need to identify the most similar pair between these two modalities as a ground-truth label. We adopt the Bipartite Matching from [19] to match the predicted 3D object boxes with the discovered novel boxes in the box pool $\mathbf{O}^{label}$. This is to guarantee that the class-specific alignment is conducted on foreground novel objects, instead of on non-object background boxes, as the object query embeddings also lead to the prediction of noisy 3D object boxes because of the limitation of supervision on novel objects during learning. If we perform class-specific alignment on noisy object boxes, it makes the 3D object feature less discriminative. When we have a matching from the box label pool $\mathbf{O}^{label}$, we then obtain the corresponding category label from the CLIP model by finding the maximum similarity score and construct a one-hot label vector $\mathbf{h}_n$ for the 3D object feature $\mathbf{F}_n^{3DObj}$ of the $n$-th object query. A detailed illustration can be seen in Fig. 2. By utilizing the discovered foreground novel object boxes, we can construct a discovery-driven cross-modal contrastive loss function as follows:

$$\mathcal{L}_{contra}^{3D\leftrightarrow Text} = \sum_{n=1}^{N} \mathbf{1}(\mathbf{F}_n^{3DObj}, \mathbf{O}^{label}) \cdot \text{CE}(\mathbf{S}_n, \mathbf{h}_n), \tag{8}$$

where $N$ is the number of object queries; $\mathbf{1}(\cdot)$ is an indicator function that returns 1 if a box matching is identified between the query and the box label pool else 0; $\text{CE}(\cdot)$ is a cross-entropy loss function. In this way, the object queries that are associated with novel objects are involved in the cross-modal alignment, leading to better feature alignment learning. The 3D features of novel objects can learn to be more discriminative, further resulting in more effective novel object discovery. In this way, the 3D novel object discovery and cross-modality feature alignment jointly boost the localization and classification for open-vocabulary 3D object detection.

## 4  Experiments

### 4.1  Experimental Setup

**Datasets and Settings.** We evaluate our proposed approach on two challenging 3D indoor detection datasets, *i.e.*, SUN-RGBD [24] and ScanNetV2 [4]. SUN-RGBD has 5,000 training samples and oriented 3D bounding box labels, including 46 object categories, each with more than 100 training samples. ScanNetV2 has 1,200 training samples containing 200 object categories [23]. Regarding the training and testing settings, we follow the strategy commonly considered in 2D open-vocabulary object detection works [8, 29]. It divides the object categories into seen and novel categories based on the number of samples of each category. Specifically, for SUN-RGBD, the categories with the top-10 most training samples are regarded as seen categories, while the other 36 categories are treated as novel categories. For ScanNetV2, the categories with the top-10 most training samples are regarded

as seen categories, other 50 categories are taken as novel categories. Regarding the evaluation metrics, we report the performance on the validation set using mean Average Precision (mAP) at an IoU threshold of 0.25, as described in [19], denoted as $AP_{25}$.

**Implementation Details.** The number of object queries for both SUN-RGBD and ScanNetv2 is set to 128, and the batch size is set to 8. We trained a base 3DETR model for 1080 epochs using only class-agnostic distillation as a base model. Subsequently, we train the base model for an additional 200 epochs with the proposed 3D novel object discovery (3D-NOD) strategy and discovery-driven cross-modal feature alignment (DCMA). Note that, to ensure a fair comparison, all the compared methods are trained using the same number of epochs. For 3D-NOD, the label pool is updated every 50 epochs. For feature alignment, considering that the discovered and GT boxes in a scene are limited, and a larger number of object queries can effectively facilitate contrastive learning, aside from the queries that match GT boxes, we choose an additional 32 object queries from the 128 queries, which are involved in the alignment with the CLIP model. The hyper-parameter settings used in the training process follow the default 3DETR configuration as described in [19].

## 4.2 Model Analysis

In this section, we provide a detailed analysis of our open-vocabulary detection framework and verify each individual contribution. The different models are evaluated on three different subsets of categories, *i.e.*, novel categories, base categories, and all the categories. These subsets are denoted as $AP_{Novel}$, $AP_{Base}$, and $AP_{Mean}$, respectively. All the evaluation metrics, including mean Average Precision (mAP) and recall (AR), are reported at an IoU threshold of 0.25. The ablation study experiments are extensively conducted on the SUN-RGBD dataset .

| Methods | $AP_{Novel}$ | $AP_{Base}$ | $AP_{Mean}$ | $AR_{Novel}$ | $AR_{Base}$ | $AR_{Mean}$ |
|---|---|---|---|---|---|---|
| 3DETR + CLIP | 3.61 | 30.56 | 9.47 | 21.47 | 63.74 | 30.66 |
| Distillation | 3.28 | 33.00 | 9.74 | 19.87 | 64.31 | 29.53 |
| 3D-NOD + Distillation | 5.48 | 32.07 | 11.26 | 33.45 | 66.03 | 40.53 |
| 3D-NOD + Distillation & PlainA | 0.85 | 35.23 | 8.32 | 34.44 | 66.11 | 41.33 |
| 3D-NOD + DCMA (full) | **6.71** | **38.72** | **13.66** | 33.66 | 66.42 | 40.78 |

Table 1: Ablation study to verify the effectiveness of the proposed 3D novel object discovery (3D-NOD) and discovery-driven cross-modality alignment (DCMA) in our 3D OV detection framework CoDA.

**Effect of class-agnostic distillation**. To demonstrate the effectiveness of class-agnostic distillation in discovery-driven alignment, we first train a class-agnostic 3DETR detection model that provides basic 3D localization for novel objects. The 3D bounding boxes are then projected onto 2D color images to crop the corresponding 2D object regions. By feeding the object regions into the CLIP model, we can conduct open-vocabulary classification to create an open-vocabulary 3D detection framework that requires both point cloud and image inputs. This baseline method is denoted as '3DETR+CLIP' in Tab. 1. We also train a 3DETR model with only the class-agnostic distillation, which serves as our base model and is denoted as 'Distillation' in Tab. 1. As shown in the table, when tested with only pure point cloud as input, the 'Distillation' model achieves comparable results to the '3DETR+CLIP' model in terms of $AP_{Novel}$ (3.28 vs. 3.61), indicating that class-agnostic distillation provides preliminary open-vocabulary detection ability for 3DETR. Moreover, given the fact that the class-agnostic distillation pushes the 3D point cloud object features and 2D image features closer wherever a 3D bounding box is predicted, and the base categories have more training object

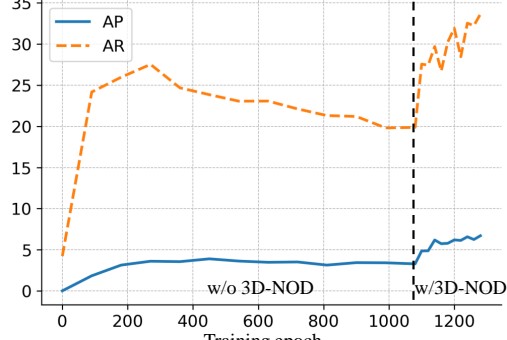

Figure 3: The influence of 3D Novel Object Discovery (3D-NOD) during training. The blue curve in the graph represents $AP_{novel}$, while the orange curve represents $AR_{novel}$. The left side of the black dashed line corresponds to the training process of our base model 'Distillation', which is trained for 1080 epochs. After that, we apply our 3D-NOD on the base model and continue to train it for an additional 200 epochs, which is represented by the right side of the black dashed line. By applying our 3D-NOD approach, we observe a significant increase in both $AP_{novel}$ and $AR_{novel}$, which breaks the limitation of base category annotations.

samples, the distillation on the base categories is more effective, leading to a higher performance in terms of $AP_{Base}$ (33.00 vs. 30.56).

**Effect of 3D Novel Object Discovery (3D-NOD).** Building upon the base model 'Distillation', we apply our 3D novel object discovery (3D-NOD) strategy to enhance the open-vocabulary detection capability, denoted as '3D-NOD+Distillation' in Tab. 1. As can be seen in the table, '3D-NOD+Distillation' achieves a higher recall than both 'Distillation' and '3DETR+CLIP' (33.45 vs. 19.87/21.47). This demonstrates that our 3D-NOD can discover more novel objects during training, leading to a higher $AP_{Novel}$ (5.48 vs. 3.28/3.61). To further analyze the influence of 3D novel object discovery during training, we monitor the entire training process and regularly evaluate the model at intermediate checkpoints. As shown in Fig. 3, the blue curve represents $AP_{novel}$ while the orange curve represents $AR_{novel}$. The left side of the black dashed line corresponds to the training process of our base model 'Distillation', which is trained for 1080 epochs. After that, we apply our 3D-NOD strategy on the base model and continue to train it for an additional 200 epochs, represented by the right side of the black dashed line. We observe that limited by the base category labels, the $AP_{novel}$ of the base model nearly stops growing at 1000 epochs. Moreover, $AR_{novel}$ begins to decrease as training continues, which meets the category forgetting phenomenon due to the lack of annotations for novel categories. However, by applying our 3D-NOD approach, we observe a significant increase in both $AP_{novel}$ and $AR_{novel}$, which demonstrates the effectiveness of our 3D-NOD approach.

**Effect of Discovery-Driven Cross-Modal Alignment (DCMA).** To explore the impact of different alignment methods on the performance of the '3D-NOD+Distillation' model. Firstly, we introduce the plain text-point alignment to the model, denoted as '3D-NOD+Distillation & PlainA'. The plain alignment involves aligning the category list of base category texts without using the discovered boxes by 3D-NOD. As shown in Tab. 1, the $AP_{Base}$ is higher than that of '3D-NOD+Distillation' (35.23 vs. 32.07), indicating that the base category texts help to improve the cross-modal feature alignment. Additionally, 3D-NOD still boosts the $AR_{Novel}$, indicating that it can still facilitate the discovery of novel objects under this comparison. However, the $AP_{Novel}$ is lower at 0.85, which shows that the model loses its discriminative ability on novel categories if without utilizing the box discovery driven by 3D-NOD. In contrast, after applying our discovery-driven class-specific cross-modal alignment method, the '3D-NOD+DCMA' model in Tab. 1 achieves the best performance on both $AP_{Novel}$ and $AP_{Base}$. This suggests that compared with the plain alignment, our discovery-driven alignment method brings more discriminative object features by using a larger category vocabulary driven by the 3D-NOD.

**Effect of the collaborative learning of 3D-NOD and DCMA.** As shown in Tab. 1, by comparing with the alignment method that is not driven by 3D-NOD (*i.e.*, '3D-NOD + DCMA' vs. '3D-NOD + Distillation & PlainA'), our 3D-NOD method improves the cross-model alignment, resulting in better performance on the novel category classification. To investigate the effect of DCMA on the novel object discovery, we apply our 3D-NOD method to the '3DETR+CLIP' pipeline without utilizing DCMA, which is denoted as '3DETR+CLIP+3D-NOD' in Tab. 2. As shown in Tab. 2, '3D-NOD+DCMA (full)' outperforms '3DETR+CLIP+3D-NOD' in terms of AP and AR for both novel and base categories. This demonstrates that the proposed DCMA can effectively help to learn more discriminative feature representations of the object query embeddings, thus improving the localization capability. Overall, these results show that 3D-NOD and DCMA together can improve the detection performance by 26% on $AP_{Novel}$ and 48% on $AP_{Base}$.

| Methods | $AP_{Novel}$ | $AP_{Base}$ | $AP_{Mean}$ | $AR_{Novel}$ | $AR_{Base}$ | $AR_{Mean}$ |
|---|---|---|---|---|---|---|
| 3DETR + CLIP [22] | 3.61 | 30.56 | 9.47 | 21.47 | 63.74 | 30.66 |
| 3DETR + CLIP + 3D-NOD | 5.30 | 26.08 | 9.82 | 32.72 | 64.43 | 39.62 |
| 3D-NOD + DCMA (full) | **6.71** | **38.72** | **13.66** | **33.66** | **66.42** | **40.78** |

Table 2: Ablation study on SUN-RGBD [24] to verify the effectiveness of the collaborative learning of DCMA and 3D-NOD. Compared with 3DETR+CLIP+3D-NOD, utilizing 3D-NOD and DCMA together can effectively improve the open-vocabulary detection performance.

**The Sensitivities of Thresholds in 3D-NOD.** In 3D novel object discovery, the utilization of the 2D semantic priors and 3D geometry priors requires two thresholds. We perform experiments to evaluate their sensitivities, and the results are presented in Tab. 3. The first row denoted by '0.0' represents the base model without our contributions. As we can see, our 3D-NOD method can produce stable

| Semantic | Geometry | $AP_{Novel}$ | $AP_{Base}$ | $AP_{Mean}$ | $AR_{Novel}$ | $AR_{Base}$ | $AR_{Mean}$ |
|---|---|---|---|---|---|---|---|
| 0.0 | 0.0 | 3.28 | 33.00 | 9.74 | 19.87 | 64.31 | 29.53 |
| 0.3 | 0.3 | **6.71** | 38.72 | **13.66** | 33.66 | 66.42 | 40.78 |
| 0.3 | 0.5 | 6.35 | **39.57** | 13.57 | 31.93 | 66.91 | 39.53 |
| 0.5 | 0.3 | 5.70 | 38.61 | 12.85 | 27.05 | 63.61 | 35.00 |
| 0.5 | 0.5 | 5.70 | 39.25 | 13.00 | 28.27 | 65.04 | 36.26 |

Table 3: Ablation study on the thresholds used for the 2D semantic and 3D geometry priors in the 3D Novel Object Discovery (3D-NOD) strategy. Disabling these priors leads to significant performance drops (see the first row), and the model achieves the best AP performance with thresholds of 0.3.

improvements with different thresholds from a wide range, demonstrating that its effectiveness is not reliant on carefully selected thresholds. As shown in Tab. 3, despite the performance variability, all our models trained with different threshold settings can consistently outperform the model trained without 3D-NOD (i.e., '0.0&0.0' in the first row) by a significant margin of 70% or more, clearly verifying its advantage. Besides, the setting with a threshold of 0.3 yields the best performance as it enables the discovery of more novel objects.

### 4.3 Comparison with Alternatives

**Quantitative Comparison.** Open-world vocabulary 3D detection is still a very new problem. Since there are very few works in the literature and our open-vocabulary setting is novel, no existing results can be directly compared with our approach. Therefore, we adapt the latest open-vocabulary point cloud classification method to our setting and evaluate its performance. Specifically, we utilize 3DETR to generate pseudo boxes and employ PointCLIP [31], PointCLIPv2 [36] and Det-CLIP[2] [30] to conduct open-vocabulary object detection, which is denoted as 'Det-PointCLIP', 'Det-PointCLIPv2' and 'Det-CLIP[2]' in Tab. 4, respectively. Additionally, we introduce a pipeline with 2D color images as input, where we use 3DETR to generate pseudo 3D boxes. We then use the camera intrinsic matrix to project 3D boxes onto 2D boxes, which are adopted to crop corresponding 2D regions. Finally, the CLIP model is utilized to classify these regions. This pipeline enables us to generate 3D detection results, denoted as '3D-CLIP' in Tab. 4. As can be observed in the table, the $AP_{Novel}$ and $AR_{Novel}$ of our method are significantly higher than other methods, further demonstrating the superiority of our design in both novel object localization and classification.

| Methods | Inputs | $AP_{Novel}$ | $AP_{Base}$ | $AP_{Mean}$ | $AR_{Novel}$ | $AR_{Base}$ | $AR_{Mean}$ |
|---|---|---|---|---|---|---|---|
| | | SUN-RGBD | | | | | |
| Det-PointCLIP [31] | Point Cloud | 0.09 | 5.04 | 1.17 | 21.98 | 65.03 | 31.33 |
| Det-PointCLIPv2 [36] | Point Cloud | 0.12 | 4.82 | 1.14 | 21.33 | 63.74 | 30.55 |
| Det-CLIP[2] [30] | Point Cloud | 0.88 | 22.74 | 5.63 | 22.21 | 65.04 | 31.52 |
| 3D-CLIP [22] | Image&Point Cloud | 3.61 | 30.56 | 9.47 | 21.47 | 63.74 | 30.66 |
| **CoDA** (Ours) | Point Cloud | **6.71** | **38.72** | **13.66** | **33.66** | **66.42** | **40.78** |
| | | ScanNetv2 | | | | | |
| Det-PointCLIP [31] | Point Cloud | 0.13 | 2.38 | 0.50 | 33.38 | 54.88 | 36.96 |
| Det-PointCLIPv2 [36] | Point Cloud | 0.13 | 1.75 | 0.40 | 32.60 | 54.52 | 36.25 |
| Det-CLIP[2] [30] | Point Cloud | 0.14 | 1.76 | 0.40 | 34.26 | 56.22 | 37.92 |
| 3D-CLIP [22] | Image&Point Cloud | 3.74 | 14.14 | 5.47 | 32.15 | 54.15 | 35.81 |
| **CoDA** (Ours) | Point Cloud | **6.54** | **21.57** | **9.04** | **43.36** | **61.00** | **46.30** |

Table 4: Comparison with other alternative methods on the SUN-RGBD and ScanNetv2 dataset. The 'inputs' in the second column refer to the testing phase. During testing, '3D-CLIP' needs 2D images as inputs to perform open-vocabulary classification for the detected objects. While our method CoDA only utilizes pure point clouds as testing inputs.

In Tab. 4, we present a comparison of our method with alternative approaches on ScanNetv2 [4]. In this evaluation, we consider the categories with the top 10 most training samples in ScanNetv2 as base categories, while the other first 50 categories presented in [23] are considered novel categories. To obtain image and point cloud pairs, we follow the approach described in [15] to generate point cloud scenes from depth maps. As shown in Tab. 4, our method significantly outperforms other approaches on AP and AR metrics for both novel and base categories, even with only point clouds as inputs during testing.

**Comparison with OV-3DET [15].** Note that the setting in our method is significantly different from OV-3DET [15]. In our setting, we train the model using annotations from a small number of base categories (10 categories) and learn to discover novel categories during training. We evaluate our model in a large number of categories (60 in ScanNet and 46 in SUN-RGBD). However, [15] relies on a large-scale pretrained 2D open vocabulary detection model, i.e., OV-2DDet model [35], to generate pseudo labels for novel categories, and evaluate the model on only 20 categories. Thus, because of the setting differences and the prior model utilized, a direct comparison with [15] is not straightforward. To provide a comparison with [15], thanks to the released codes from the authors of [15] to generate pseudo labels on ScanNet, we can directly retrain our method following the same setting of [15] on ScanNet, and provide a fair comparison with [15] on Tab. 5. As can be observed, our method achieves clearly better mean AP (1.3 points improvement over all the categories), validating the superiority of our method.

| Methods | Mean | toilet | bed | chair | sofa | dresser | table | cabinet | bookshelf | pillow | sink |
|---|---|---|---|---|---|---|---|---|---|---|---|
| OV-3DET [15] | 18.02 | 57.29 | 42.26 | 27.06 | 31.50 | 8.21 | 14.17 | 2.98 | 5.56 | 23.00 | 31.60 |
| **CoDA** (Ours) | **19.32** | 68.09 | 44.04 | 28.72 | 44.57 | 3.41 | 20.23 | 5.32 | 0.03 | 27.95 | 45.26 |

| Methods | | bathtub | refrigerator | desk | nightstand | counter | door | curtain | box | lamp | bag |
|---|---|---|---|---|---|---|---|---|---|---|---|
| OV-3DET [15] | | 56.28 | 10.99 | 19.72 | 0.77 | 0.31 | 9.59 | 10.53 | 3.78 | 2.11 | 2.71 |
| **CoDA** (Ours) | | 50.51 | 6.55 | 12.42 | 15.15 | 0.68 | 7.95 | 0.01 | 2.94 | 0.51 | 2.02 |

Table 5: Comparison of methods in the same setting of [15] on ScanNet. 'Mean' represents the average value of all the 20 categories.

**Qualitative Comparison.** As depicted in Fig. 4, our method demonstrates the strong capability to discover more novel objects from point clouds. For instance, our method successfully identify the dresser in the first scene and the nightstand in the second scene. The performance comparison indicates that our method generally exhibits superior open-vocabulary detection ability on novel objects.

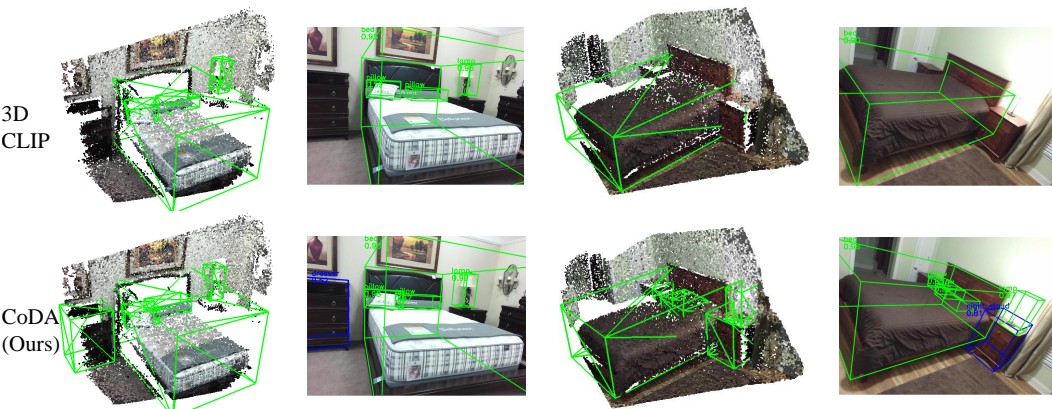

Figure 4: Qualitative comparison with 3D-CLIP [22]. Benefiting from our contributions, our method **CoDA** can discover more novel objects, which are indicated by blue boxes in the color images.

## 5   Conclusion

This paper proposes a unified framework named CoDA to address the fundamental challenges in OV-3DDet, which involve localizing and classifying novel objects. To achieve 3D novel object localization, we introduce the 3D Novel Object Discovery strategy (3D-NOD), which employs both 3D box geometry priors and 2D semantic open-vocabulary priors to discover more novel objects during training. To classify novel object boxes, we further design a Discovery-driven Cross-Modal Alignment module (DCMA) that includes both a class-agnostic and a class-discriminative alignment to align features of 3D, 2D and text modalities. Our method significantly outperforms alternative methods by more than 80% in mAP.

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
