# CoDA: Collaborative Novel Box Discovery and Cross-modal Alignment for Open-vocabulary 3D Object Detection
## – Supplemental Material –

# 1   More Ablation Studies

## 1.1   Number of test categories in evaluation

In the ablation studies of our main paper, we evaluate our method on SUN-RGBD [8] with 46 classes, including 36 novel classes. To further explore the open-vocabulary abilities of our method, we expand the number of test categories from 40 to 200 to evaluate our model. As illustrated in Fig. A, as the number of test categories increases, $AP_{Mean}$ and $AP_{Novel}$ decrease gradually to 4.68% and 2.85%, respectively. Considering that the inputs are pure point clouds, which are less discriminative than color images, the open-vocabulary ability remains limited when more categories are introduced, which is a general challenge of related 3D open-vocabulary scene understanding works [6, 2, 3]. In future work, we may focus on incorporating multi-modality inputs to further improve the performance on the large vocabularies.

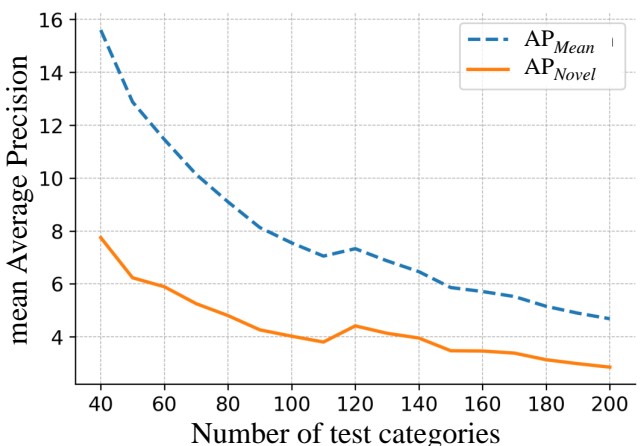

Figure A: AP results on SUN-RGBD [8] with larger numbers of test categories (40~200). '$AP_{Novel}$' denotes mAP on novel categories, and '$AP_{Mean}$' denotes mAP on all the categories (base and novel categories). The open-vocabulary capability remains limited when significantly more novel vocabularies are introduced, considering pure point clouds are less discriminative than color images, which is a serious challenge in 3D open-vocabulary scene understanding [6, 2, 3].

## 1.2 Class-agnostic distillation vs. Class-specific distillation

(i) When we **do not apply 3D-NOD**, the comparison between the class-agnostic and class-specific distillation is shown in Tab. A. As discussed in Sec 3.3 of the main paper, the key difference is whether the method considers the category labels for GT 3D boxes when constructing the distillation objective. In Tab. A, the $AP_{Novel}$ of 'Class-agnostic Distillation' outperforms 'Class-specific Distillation', further proving that the class-agnostic distillation offers better generalization for novel categories.

| Methods | $AP_{Novel}$ | $AP_{Base}$ | $AP_{Mean}$ | $AR_{Novel}$ | $AR_{Base}$ | $AR_{Mean}$ |
|---|---|---|---|---|---|---|
| Class-specific Distillation | 1.16 | 33.91 | 8.28 | 19.92 | 63.95 | 29.49 |
| Class-agnostic Distillation | 3.28 | 33.00 | 9.74 | 19.87 | 64.31 | 29.53 |

Table A: Comparison between class-specific distillation and class-agnostic distillation without 3D-NOD.

(ii) When we **apply 3D-NOD**, the class-specific contrastive learning [4] encourages features belonging to the same categories to be as similar as possible. However, during the discovery process, the discovered boxes may cover background regions. Since CLIP cannot classify the background, boxes on background with higher scores may be assigned to incorrect categories, thus leading to severe feature misalignment by class-specific contrastive learning. Conversely, our class-agnostic distillation aims at making the 3D object query features align with their corresponding projected 2D image features. The distillation can also cover 3D object boxes on the background. So the distillation can be effectively performed for the foreground and background simultaneously. To show our advantage compared to class-specific contrastive learning, we replace class-agnostic distillation with class-specific contrastive learning in our method. As shown in Tab. B, the contrastive loss encourages more boxes to be classified as the foreground, resulting in a larger recall. However, as discussed, it also encourages more background boxes to be misclassified as foreground objects, leading to a lower overall mAP compared to our class-agnostic distillation.

| Methods | $AP_{Novel}$ | $AP_{Base}$ | $AP_{Mean}$ | $AR_{Novel}$ | $AR_{Base}$ | $AR_{Mean}$ |
|---|---|---|---|---|---|---|
| 3D-NOD + DCMA (contrastive loss) | 5.35 | 34.58 | 11.70 | 37.93 | 66.93 | 44.24 |
| 3D-NOD + DCMA (class-agnostic distillation) | **6.71** | **38.72** | **13.66** | 33.66 | 66.42 | 40.78 |

Table B: Comparison between class-specific and class-agnostic distillation with 3D-NOD.

## 1.3 The influence of CLIP on the performances

(i) When we **do not consider novel categories** and train our model in the close-vocabulary setting in 3DETR [5] (*i.e.*, keeping the cross-modal alignments and discarding the 3D-NOD), CLIP does not degrade the performance as shown in Tab. C. This is because, during the training, the CLIP knowledge can be adapted for 3D detection on the base categories by learning the cross-modal alignment.

| Methods | **Mean** | chair | table | desk | bed | sofa | toilet | dresser | nightstand | bookshelf | bathtub |
|---|---|---|---|---|---|---|---|---|---|---|---|
| 3DETR [5] | 56.80 | 68.00 | 50.00 | 28.70 | 81.80 | 58.30 | 90.30 | 28.60 | 56.60 | 27.50 | 77.60 |
| **CoDA** (Ours) | **57.98** | 69.33 | 49.01 | 28.79 | 85.54 | 62.88 | 91.61 | 29.65 | 61.89 | 25.88 | 75.27 |

Table C: Performance comparison with the 3DETR [5] in the same close-vocabulary 3DETR settings.

(ii) When we **consider novel categories and directly apply pre-trained CLIP in the testing stage** to achieve open-vocabulary detection (*i.e.*, '3D-CLIP', the open-vocabulary baseline model), the mAP of the base categories of '3D-CLIP' is lower than the AP of 3DETR, as shown in Tab. D. The base categories of our method in this experiment are the same as the categories in the 3DETR paper. It is reasonable because the pre-trained CLIP is directly used for open-vocabulary classification in the testing. There is no cross-modal alignment between CLIP and the target data distribution during training. However, '3D-CLIP' can detect objects of novel categories and obtain an $AP_{Novel}$ of 2.39, while the 3DETR cannot perform novel object detection, further confirming the benefits of using CLIP for the problem.

(iii) When we **consider novel categories and involve CLIP during training in our designed method**, the $AP_{Novel}$ and $AP_{Base}$ of our method are both significantly higher than the open-vocabulary baseline model '3D-CLIP', as shown in Tab. D. While $AP_{Base}$ of our method is slightly

| Methods | $AP_{Novel}$ | $AP_{Base}$ |
|---|---|---|
| 3DETR [5] | - | 56.80 |
| 3D-CLIP [7] | 2.39 | 41.64 |
| **CoDA** (Ours) | 5.13 | 55.26 |

Table D: Performance comparison using the same 10 categories in the 3DETR paper as the base categories. '3DETR' results show the performance of the original camera-ready version of 3DETR [5]. '-' indicates that the close-vocabulary 3DETR cannot perform detection on novel categories. '3D-CLIP' is our open-vocabulary baseline model. Our method CoDA achieves significant improvements on both $AP_{Novel}$ and $AP_{Base}$ upon the baseline.

lower than 3DETR, we think it is a reasonable phenomenon since 3D-NOD may also possibly discover noisy 3D boxes and the method needs to jointly learn both the base and the novel categories which are more challenging compared to 3DETR [5] that only learns the base categories.

## 1.4 Other class-splitting settings

When we set the same 10 categories of 3DETR as our base categories. As shown in Tab. D, our method also achieves significant improvements on both $AP_{Novel}$ and $AP_{Base}$ compared to the open-vocabulary baseline model, further demonstrating the effectiveness of our proposed method.

## 1.5 Upper bound of training a model on all the categories

To verify the performance upper bound, we train the model with both the base and novel annotations, as shown in Tab. E, while our method demonstrates significant improvements in 3D novel object detection, compared with the upper bound, there is still room for further investigation.

| Methods | $AP_{Novel}$ | $AP_{Base}$ | $AP_{Mean}$ | $AR_{Novel}$ | $AR_{Base}$ | $AR_{Mean}$ |
|---|---|---|---|---|---|---|
| CoDA (Ours) | 6.71 | 38.72 | 13.66 | 33.66 | 66.42 | 40.78 |
| Supervised by Base+novel GT | 13.45 | 29.71 | 16.99 | 50.10 | 67.04 | 53.78 |

Table E: The upper bound of adopting base and novel annotations in SUN-RGBD.

## 1.6 Cross-dataset evaluation

The comparisons between the cross-dataset and the same-dataset performances in terms of $AP_{Novel}$ are 3.88 vs. 6.54 on ScanNet [1] and 2.64 vs. 6.71 on SUN-RGBD. The lower performances of the cross-dataset evaluation can be due to the data domain gap from the utilization of different RGB-D sensors.

## 1.7 Model complexity discussion

During testing, our method only adds the CLIP-text encoder and one more distillation head to 3DETR. Therefore, the parameters are just slightly more than 3DETR [19] backbone (6.99M vs. 5.81M), demonstrating that our designs will not bring many extra computations.

## 1.8 Visualization for detecting objects beyond the 200 categories

As shown in Fig. C, our method can detect the objects belonging to the 'folder' category and beyond the 200 categories in ScanNet-200, which are labeled by blue boxes in the color image of the second row. It shows that our method has the potential to extend the application to larger vocabularies.

## 1.9 Failure cases

As shown in Fig. D, failure cases in challenging scenes can be observed for similar semantic categories. For instance, in the second row, a 'coffee table' is misclassified as a 'table'. This issue may be further improved by introducing a stronger classifier than CLIP in future work.

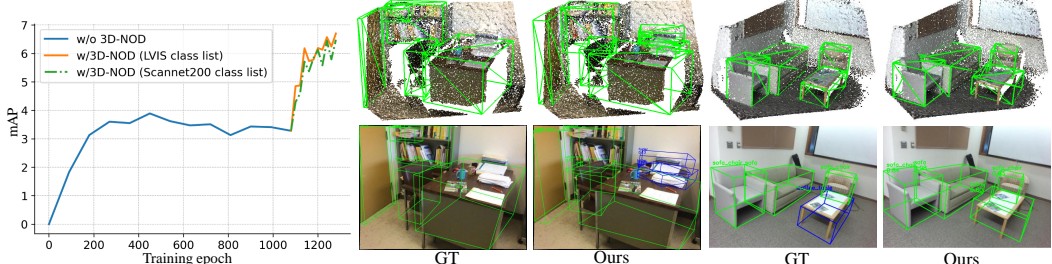

Figure B: The blue curve represents $AP_{novel}$ of the training process of our base model 'Distillation', which is trained for 1080 epochs. After that, we apply our 3D-NOD on the base model and continue to train it for an additional 200 epochs. The orange curve represents $AP_{novel}$ of the model trained by 3D-NOD with the LVIS class list. The green curve represents $AP_{novel}$ of the model trained by 3D-NOD with the ScanNet200 class list.

Figure C: Detection beyond 200 categories. The detected objects indicated by blue boxes in the color image (the second row), belong to the category 'folder', which is not in the 200 categories in ScanNet-200, showing that our method has the capabilities to be applied to a larger vocabulary.

Figure D: Qualitative illustration of failure cases. Similar semantic categories may lead to incorrect GT classification results. For instance, in the second row, a GT 'coffee table' with a blue box is misclassified as a 'table'.

## 1.10 The convergence rate when adopting the ScanNet-200 category list

We conduct experiments on SUN-RGBD with the categories of ScanNet200 as the super-category list for 3D-NOD. As shown in Fig. B, the training process of our base model, denoted as 'Distillation', is corresponding to the blue curve. The orange curve represents the $AP_{novel}$ of the model trained using 3D-NOD with the LVIS class list, while the green curve indicates the model trained using 3D-NOD with the ScanNet200 class list. From the figure, we can observe that the convergence rate is not significantly affected when replacing the category list with the ScanNet200 list. Notably, the green curve is slightly lower than the orange curve, which is probably because of the significant change in the number of classes, *i.e.*, from a small number of classes (200 categories) to a much larger number of classes (1.2k categories) in the LVIS class list.

## 1.11 Comparison with a fully supervised ScanNet-200 method

We trained a fully supervised model on ScanNet-200. The mAP achieved for all categories is 9.66. Our proposed method achieved mAP of 2.74 across all the categories. As discussed in Sec. 1.1, the open-vocabulary 3D detection ability of our approach remains limited when evaluated on a larger number of categories (like 200 categories) despite a state-of-the-art performance.

## 1.12 Ablation study for 3D-NOD and DCMA on ScanNet

We perform our ablation on the SUN-RGBD dataset in our main paper. This choice was made because SUN-RGBD also includes 200+ categories in the released official human annotation, and the training time on SUN-RGBD is shorter compared to ScanNet due to the smaller amount of training data. To further demonstrate that the conclusions from the ablation studies are consistent between SUN-RGBD and ScanNet, we also conducted the ablation study in Tab. 1 of the main paper on ScanNet in this supplemental material. The results are shown in Tab. F. Both Tab. 1 in the main paper and Tab. F indicate that our 3D-NOD and DCMA methods significantly improve performance over the baseline, further confirming the effectiveness of the proposed method.

## 1.13 Effect of Different Matching Methods in 3D-NOD

In the discovery-driven alignment, we need to match predicted boxes with boxes in the discovered box label pool to construct the contrastive alignment loss. The matched predicted boxes are used in the contrastive feature alignment, as shown in Fig.2 of the main paper. We thus compare different matching methods, and show the results of the 3DIoU match and the Bipartite match. For the 3DIoU

| Methods | $AP_{Novel}$ | $AP_{Base}$ | $AP_{Mean}$ | $AR_{Novel}$ | $AR_{Base}$ | $AR_{Mean}$ |
|---|---|---|---|---|---|---|
| 3DETR+CLIP | 3.74 | 14.14 | 5.47 | 32.15 | 54.15 | 35.81 |
| Distillation | 2.91 | 20.50 | 5.84 | 34.40 | 59.63 | 38.61 |
| 3D-NOD+Distillation | 4.25 | 13.74 | 5.84 | 49.13 | 55.08 | 50.12 |
| 3D-NOD+Distillation & PlainA | 0.50 | 12.90 | 2.56 | 44.14 | 54.00 | 45.79 |
| 3D-NOD + DCMA (full) | **6.54** | **21.57** | **9.04** | 43.36 | 61.00 | 46.30 |

Table F: Ablation study on ScanNet to verify the effectiveness of the proposed 3D novel object discovery (3DNOD) and discovery-driven cross-modality alignment (DCMA) in our 3D OV detection framework.

| Methods | $AP_{Novel}$ | $AP_{Base}$ | $AP_{Mean}$ | $AR_{Novel}$ | $AR_{Base}$ | $AR_{Mean}$ |
|---|---|---|---|---|---|---|
| 3D-NOD+Distillation | 5.48 | 32.07 | 11.26 | 33.45 | 66.03 | 40.53 |
| 3DIoU | 5.68 | 31.29 | 11.25 | 33.08 | 67.35 | 40.53 |
| Bipartite | **6.71** | **38.72** | **13.66** | 33.66 | 66.42 | 40.78 |

Table G: Comparison of matching methods for the 3D Novel Object Discovery (3D-NOD) strategy.

match, we activate the predicted 3D boxes if the boxes have 3DIoU value larger than 0.25 with any box in the label pool. As shown in Tab. G, both the $AP_{Novel}$ and $AP_{Base}$ of the 3DIoU matching are lower than 'Bipartite' and comparable with '3D-NOD+Distillation', which indicates that the 3DIoU matching method does not work well. The reason behind this is that indoor scenes are denser compared to outdoor scenes. Thus, if the predicted box has a shift, the object it covers may change. While 3DIoU matching assigns multiple predicted boxes for one 3D ground-truth box if their 3DIoU is larger than 0.25, leading to semantic confusion and invalidity for the alignment. Therefore, the Bipartite matching is preferred over the 3DIoU matching in our discovery-driven alignment.

## 2 Comparison with Alternative

### 2.1 Qualitative Comparison

In Fig. E and Fig. F, we show more detection results from different scenes. Compared with '3D-CLIP' [7], our models have better results on both novel categories (blue boxes in 2D color images) and base categories (green boxes in 2D color images), further confirming the effectiveness of the proposed collaborative learning of the designed 3D-NOD and DCMA strategies.

## 3 Limitations

As discussed in Sec. 1.1, considering the pure point clouds are less discriminative than color images, the open-vocabulary detection ability for larger vocabulary is still limited, which is a common challenge for related 3D open-vocabulary scene understanding [6, 2, 3]. In the future, we may introduce multi-modality inputs to improve the performance on the large vocabulary.

## 4 Potential negative societal impact

In our work, we train our method only on public datasets [8, 1], ensuring that no private data is utilized. After releasing the codes and pre-trained models, some people may possibly consider using our method for large-scale open-world training involving personal data, causing the data privacy issue. We will explicitly cover this issue when we release the project and restrict any utilization of our project that may cause a potential negative societal impact.

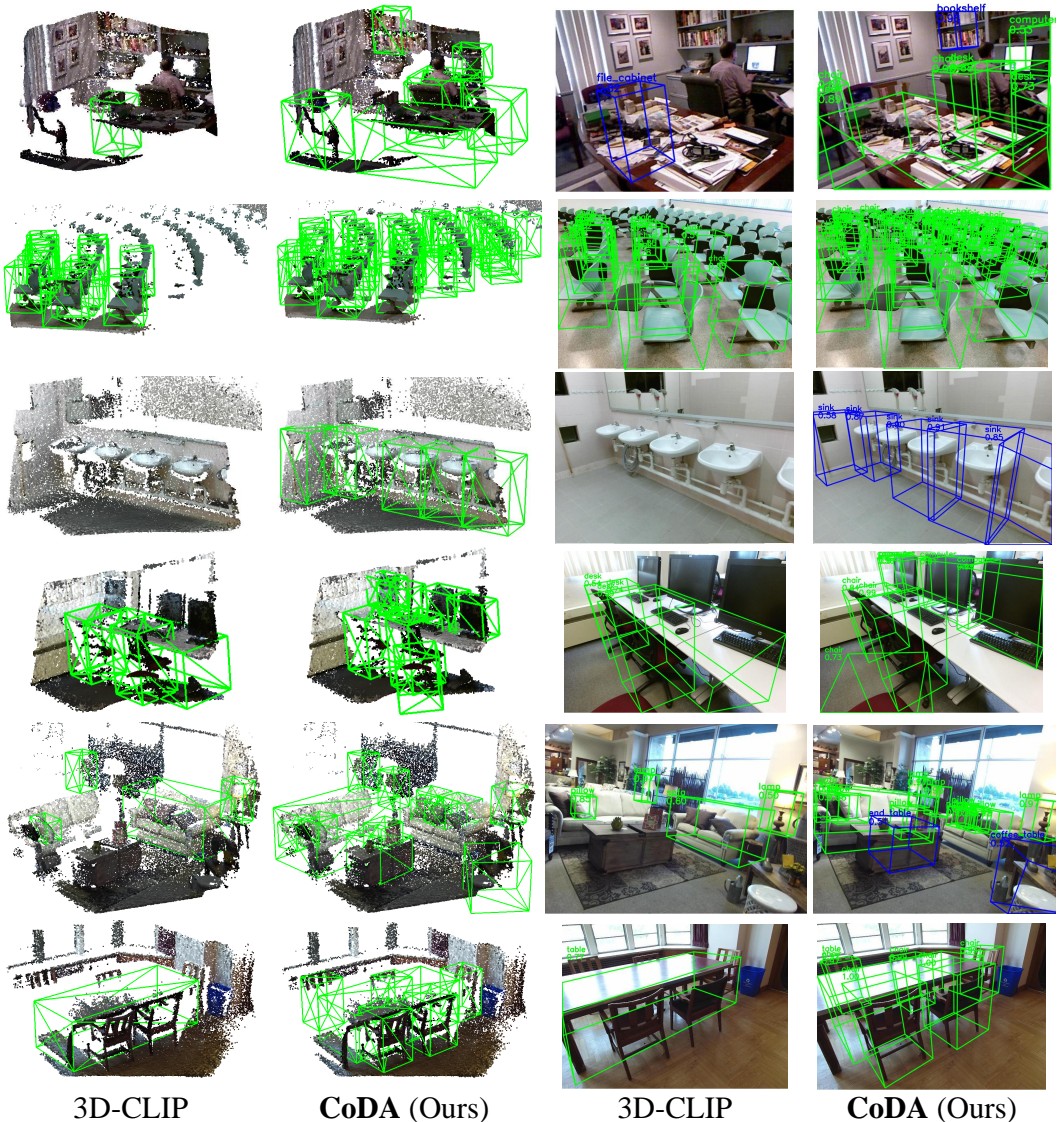

| 3D-CLIP | **CoDA** (Ours) | 3D-CLIP | **CoDA** (Ours) |

Figure E: Qualitative comparison with 3D-CLIP [7]. Benefiting from our contributions, our method can discover more novel objects, which are indicated by blue boxes in the color images. Besides, our method can also detect more base objects, which proves that our method has better open-world detection capabilities, with the proposed collaborative 3D-NOD and Cross-modal Alignment. Here only the objects with classification scores larger than 0.5 are displayed for better visualization.

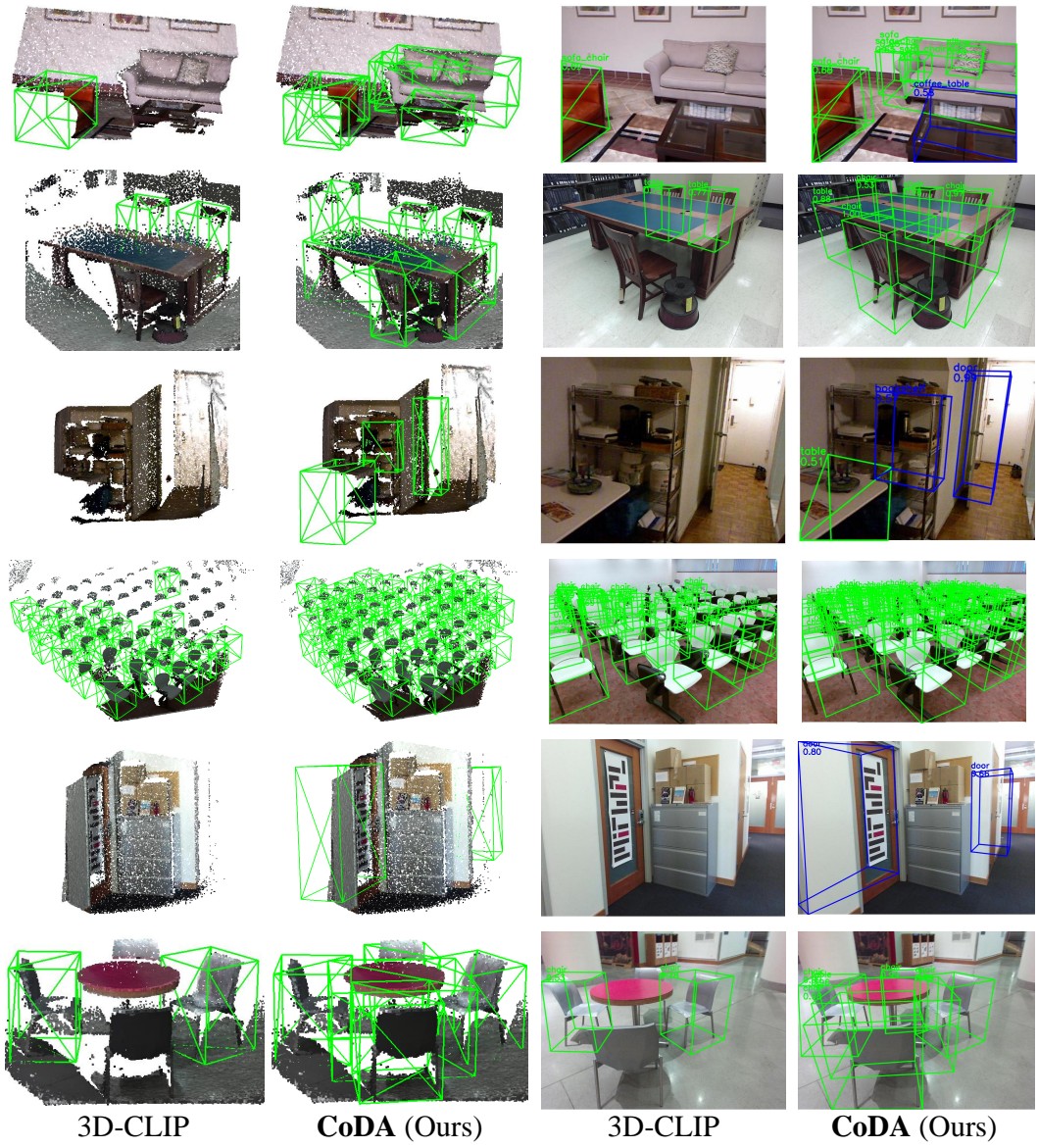

| 3D-CLIP | **CoDA** (Ours) | 3D-CLIP | **CoDA** (Ours) |

Figure F: Qualitative comparison with 3D-CLIP [7]. Benefiting from our contributions, our method can discover more novel objects, which are indicated by blue boxes in the color images. Besides, our method can also detect more base objects, which proves that our method has better open-world detection capabilities, with the proposed collaborative 3D-NOD and Cross-modal Alignment. Here only the objects with classification scores larger than 0.5 are displayed for better visualization.