# OpenReview forum: "CoDA: Collaborative Novel Box Discovery and Cross-modal Alignment for Open-vocabulary 3D Object Detection"
_NeurIPS.cc/2023/Conference — NeurIPS 2023 poster_

### Official Review · Reviewer_goao · 2023-06-29

**Soundness:** 2 fair
**Presentation:** 2 fair
**Contribution:** 2 fair
**Rating:** 5
**Confidence:** 5

**Summary:**

This paper addresses the challenge of open-vocabulary 3D object detection (OV-3DDet). The proposed framework consists of two modules: the 3D Novel Object Discovery (3D-NOD) module and the Discovery-Driven Cross-Modal Alignment (DCMA) module. The 3D-NOD module utilizes a detector trained on annotations of base categories (without class-specific information) to localize potential objects. It then employs the pre-trained CLIP to discover the potential super categories of these objects. In the DCMA module, two types of alignments are performed. First, the 3D object features are aligned with their corresponding 2D object features using the CLIP image encoder. Second, the 3D object features are aligned with their corresponding text features using the CLIP text encoder. The proposed framework is mainly evaluated on the SUN RGB-D dataset to assess its performance and effectiveness.

**Strengths:**

1. The problem of open-vocabulary 3D object detection addressed in this paper is highly relevant in real-world scenarios where the number of object classes is not predefined or limited. The ability to detect and classify objects in an open-world setting is crucial for applications such as autonomous driving, robotics, and augmented reality.

2. The proposed method is technically sound. The approach of jointly training the novel object discovery module for localization and the cross-modal alignment module for classification is well-motivated.

**Weaknesses:**

1. The experimental results presented in the paper lack convincing evidence and could be improved in several aspects.

+ First, in Table 4, it would be beneficial to compare the proposed method with the existing method [14], which is the first work on OV-3DDet, as well as [R1], which explores both open-vocabulary recognition and open-vocabulary localization. This would provide a clear assessment of the effectiveness of the proposed approach. Additionally, it is mentioned in line 150 and Figure 2 that the proposed method utilizes both images and point clouds, making an incorrect comparison with [22] in terms of inputs.

+ Second, in Table 1, the detailed settings of the compared methods, such as distillation and 3D-NOD + Distillation, are unclear. It is essential to clarify how super category annotations for novel objects are obtained without 3D-NOD. Furthermore, it is unclear whether the compared methods are trained with the same number of epochs, which can significantly impact the performance comparison.

+ Third, all experiments in the main paper are conducted solely on the SUN RGB-D dataset, it would be valuable to include evaluation results on the ScanNetV2 dataset,  which contains much more categories and is a more practical setting for open-vocabulary learning. Moreover, it is mentioned in line 227-228 that the remaining categories are treated as novel on the both datasets, indicating the presence of 190 novel categories on ScanNetV2. However, Table B in the appendix only considers 50 categories on ScanNetV2 as novel categories, creating a discrepancy.

+ Lastly, according to Table 3, the performance seem to be sensitive to the two hyperparameters. It would be beneficial to provide more insights and analysis regarding the sensitivity of these hyperparameters.

2. The paper's writing style tends to be wordy, leading to repeated chunks of information in the introduction, figure captions, and methodology sections. Streamlining the text and avoiding redundancy would enhance the overall readability and clarity of the paper.

[R1] Zeng, Yihan, et al. "CLIP2: Contrastive Language-Image-Point Pretraining from Real-World Point Cloud Data." Proceedings of the IEEE/CVF Conference on Computer Vision and Pattern Recognition. 2023.

**Questions:**

1. The correctness of Eq. 4 is questionable. It is unclear why a novel object must have an IoU over 0.25 with objects from base categories. Further clarification and justification from the authors would be helpful to understand the rationale behind this threshold.

2. It is not explicitly mentioned whether the base objects are included in O^novel. If the base objects are not part of O^novel, it is important to clarify how the base objects contribute to the cross-modal alignment when only O^novel is used in computing the alignment loss (Eq. 8).

3. In line 235, the meaning of "additional 32 object queries" is not explicitly explained. It is unclear whether it refers to the total number of queries in this alignment being 128+32.

**Limitations:**

The authors did not discuss its limitations and potential negative societal impact of their work.

---

> ### Author Rebuttal · Authors · 2023-08-10
>
> **Q1 Comparison with existing method [15] and [R1]:**
>
> Regarding the comparison with [15], please refer to Tab. B of the rebuttal PDF. As for the comparison with [R1] (i.e., CLIP$^2$), we conducted comparison experiments on SUN-RGBD. More specifically, we utilize 3DETR to generate pseudo boxes and employ CLIP$^2$ to conduct open-vocabulary object detection, which is denoted as 'Det-CLIP$^2$'. As shown in Tab. F in the rebuttal PDF and Tab. 4 in the main paper, with pure point clouds as the testing input, Det-CLIP$^2$ [R1] achieves better AP than Det-PointCLIP [30] and Det-PointCLIPv2 [35], while our method still significantly outperforms them.
>
> **Q2 The comparison with [22] in terms of inputs:**
>
> The inputs we discussed here refer to the testing phase, instead of the training phase. During testing, the method [22] needs 2D images as inputs to perform open-vocabulary classification for the detected objects. While our method only utilizes pure point clouds as testing inputs. This is made possible because DCMA aligns the 3D features with 2D image and text features during training. Thus, the comparison with [22] in terms of inputs is correct. We will update a clearer explanation.
>
> **Q3 How super category annotations for novel objects are obtained without 3D-NOD:**
>
> As discussed in L177-L187 of the main paper, class-agnostic distillation eliminates the requirement of category labels of GT boxes. Thus, the training for our 'Distillation' does not need category annotations for novel objects. We will further clarify this point in the revision.
>
> **Q4 Whether the compared methods are trained with the same number of epochs:**
>
> To ensure a fair comparison, they are trained using the same number of epochs. We will clearly describe this setting in the revision. We will also release all the codes and pre-trained models once the paper is accepted.
>
> **Q5 It would be valuable to include evaluation results on the ScanNet:**
>
> We have included comparisons with alternative methods on the ScanNet, as presented in Tab. B of the supplementary material. Additionally, in this rebuttal, we compared our methods on ScanNet with the latest OV-3DDet work [15] following their data and experimental setups, and the results are shown in Tab. B of the rebuttal PDF. We achieve better performances (1.3 points mAP improvement) compared to [15].
> Regarding the ablation studies, we performed them on the SUN-RGBD dataset. This choice was made because SUN-RGBD also includes 200+ categories in the released official human annotation, and the training time on SUN-RGBD is shorter compared to ScanNet due to the smaller amount of training data. To further demonstrate that the conclusions from the ablation studies are consistent between SUN-RGBD and ScanNet, we also conducted the ablation study in Tab. 1 of the main paper on ScanNet in this rebuttal. The results are shown in Tab. G of the rebuttal PDF. Both Tab. G and Tab. 1 in the main paper indicate that our 3D-NOD and DCMA methods significantly improve performance over the baseline, further confirming the effectiveness of the proposed method.
>
> Thanks for your suggestion. We will merge these results above into the updated version.
>
> **Q6 Unclear statement for the setting in ScanNet:**
>
> Sorry for the unclear statement and thanks for pointing this out. To clarify, in the settings of ScanNet, the methods are evaluated on 60 categories (i.e., 10 base + 50 novel). Regarding the ablation study on the larger vocabulary (i.e., 200 categories), as shown in Fig. A in the Supplementary, we conducted it on the SUN-RGBD dataset as the other ablation study. We will make careful revisions.
>
> **Q7 Sensitivity to the two hyperparameters:**
>
> In Tab. 3 of our main paper, we conducted an ablation study on the thresholds for the 2D semantic and 3D geometry priors in 3D-NOD. The entry '0.0&0.0' in the first row represents that the model is trained without 3D-NOD, which is our base model. When incorporating 3D-NOD with different thresholds for semantic and geometric information, all the trained models with different thresholds can consistently outperform '0.0&0.0' by a significant margin of 70% or more. This demonstrates that the effect of 3D-NOD is not sensitive to these hyperparameters.  Interestingly, the '0.3&0.3' threshold achieved the highest $AP_{Novel}$. This is because a lower threshold in 3D-NOD leads to the discovery of more novel objects, aligning with our initial intuition.
>
> **Q8 Writing style:**
>
> We will carefully polish the paper in the revision.
>
> **Q9 Clarification for the threshold:**
>
> Apologies for this typo and thanks for pointing this out. In Eq.4 of our main paper, the condition should be IoU < 0.25. This ensures that the discovered boxes do not overlap too much with the base objects, thereby guaranteeing as much as possible that they represent novel objects. We will revise accordingly for this in the updated version.
>
> **Q10 If only $O^{novel}$ is used in computing the alignment loss:**
>
> Both the base and novel objects contribute to the cross-model alignment in our method. The symbol $O^{novel}$ in Eq. 8 should be $O^{label}$. We will update it in the revision.
>
> **Q11 Unclear description of the additional 32 object queries:**
>
> Please refer to the answer for Q6 of Reviewer Qxx5.
>
> **Q12 Limitation:**
>
> Please refer to the answer for Q6 of Reviewer zhog.
>
>
> [R1] Zeng, Yihan, et al. "CLIP2: Contrastive Language-Image-Point Pretraining from Real-World Point Cloud Data." Proceedings of the IEEE/CVF Conference on Computer Vision and Pattern Recognition. 2023.

---

> > ### Comment · Reviewer_goao · 2023-08-17
> > **Post-rebuttal Comment**
> >
> > Thank the authors for their thorough response, the clarifications provided, and the additional results. Many of my concerns have been addressed. However, I have two more questions:
> >
> > 1. Why wasn't there a comparison with [15] on the 20 common classes on SUN RGB-D?
> > 2. The response to Q7 still lacks persuasiveness in my view. It seems evident that varying thresholds in 3D-NOD lead to distinct performance differences. How can the authors assert that "This demonstrates that the effect of 3D-NOD is not sensitive to these hyperparameters"?

---

> > > ### Author Response · Authors · 2023-08-18
> > >
> > > Dear Reviewer goao, thank you very much for your careful reading of the rebuttal, and we sincerely appreciate your further reply. We provide the following answers to your further questions:
> > >
> > > **Q1 Comparison with [15] on the 20 common classes on SUN RGB-D:**
> > >
> > > We would like to clarify the **significant differences in the settings** between ours and [15]. In [15], they adopt an extra open-vocabulary 2D detection model [34] to generate large-vocabulary 3d pseudo labels on 1k categories. Then their model is trained by these large-vocabulary 3d pseudo labels. While in our settings, rather than using any open-vocabulary 2D detection model, we train a base model with 3D labels of base categories (10 categories). Then we discover more and more objects of novel categories during training. As the distinct difference in the task settings, it is not straightforward to directly compare our model with theirs. In order to provide a fair comparison with [15], we need to adapt our model to their setting. Specifically, we train our base model by **the same 3d pseudo labels** with [15], then discover more and more objects during training. We appreciate the authors of [15] for releasing their codes for generating pseudo labels on ScanNet. This enables us to train our method using the same pseudo labels and evaluate our model under the same setting. As shown in Tab. B of the rebuttal PDF, our method achieves **clearly better mean AP (1.3 points improvement)**, proving the superiority of our method. While the codes of [15] for generating 3D pseudo labels **for SUN-RGBD are not released yet** at the time of our reply. Additionally, the ScanNet and SUN-RGBD are captured using different sensors, which introduces significant gaps in the data and processing for the pseudo-label generation. Thus, the codes for ScanNet cannot be directly transferred to SUN-RGBD without detailed descriptions. To ensure fairness and correctness, we only compare our method on ScanNet for the present. Considering that (i) both our model and [15] do not have dataset-specific designs for either ScanNet or SUN-RGBD, and (ii) our analysis in response to Q5 of the rebuttal highlights the consistency of our method across the two datasets. The already provided comparison on ScanNet can **effectively validate the superiority of our methods**. We will also contact the authors of [15] for the remaining parts of source codes and update the comparison on SUN-RGBD in the final paper.
> > >
> > > **Q2 The sensitivity of these hyperparameters:**
> > >
> > > We apologize for the confusion caused. We wanted to express that our 3D-NOD method can produce stable improvements with different thresholds from a wide range, demonstrating that its effectiveness is not reliant on carefully selected thresholds. As shown in Tab. 3 in the main paper, despite the performance variability, all our models trained with different threshold settings can consistently outperform the model trained without 3D-NOD (*i.e.*, ‘0.0&0.0’ in the first row of Tab. 3) by a significant margin of 70% or more, clearly verifying its advantage. We will further revise our statement about the sensitivity in the revision to make this point clearer.

---

> > > > ### Author Response · Authors · 2023-08-21
> > > >
> > > > Dear Reviewer goao, thank you very much for providing your valuable comments. As the deadline is approaching, we would like to kindly inquire if our discussions have addressed your concerns. If you have any further questions, we would be happy to continue our conversation. If our response has cleared your concerns, we would greatly appreciate it if you could consider updating your score and leaving your feedback. Thank you again.

---

### Official Review · Reviewer_zhog · 2023-07-06

**Soundness:** 3 good
**Presentation:** 3 good
**Contribution:** 3 good
**Rating:** 6
**Confidence:** 3

**Summary:**

This paper aims to address the problem of detecting objects from an arbitrary list of categories within a 3D scene, which is less explored area in the literature. It proposes a unified framework that simultaneously addresses the fundamental problems of localizing and classifying novel objects, under the condition of limited base categories. The framework incorporates a 3D Novel Object Discovery (3D-NOD) strategy to localize novel 3D objects, which utilizes both 3D box geometry priors and 2D semantic open-vocabulary priors to generate pseudo box labels of the novel objects. Besides, it introduces a cross-modal alignment module based on discovered novel boxes, to align feature spaces between 3D point cloud and image/text modalities for novel object box classification. Experiments on the SUN-RGBD and ScanNet demonstrate the effectiveness.

**Strengths:**

1. The idea makes sense to me. The framework includes a system that discovers new 3D objects and aligns them with other types of data. This helps the system learn better and eliminates the need for a 2D detector. The system also uses CLIP semantics to find objects in 3D scenes during training.
2. The proposed framework performs much better than other methods on the ScanNet-200 and SUN RGBD datasets and achieves a significant improvement in mAP.

**Weaknesses:**

1. The paper doesn't discuss how the proposed framework's computational complexity compares to other state-of-the-art methods, or how efficient it is.
2. The paper doesn't include an analysis of cases where the proposed framework fails to detect objects.
3. It would be helpful to analyze how the convergence rate is affected when the novel object list is similar to the ScanNet-200 dataset.
4. The experiments don't compare the proposed framework to a fully supervised ScanNet-200 method, which would help readers understand the gap in detecting novel objects. Additionally, comparing the proposed framework to an oracle-supervised method with the same model would be useful in understanding the usefulness and importance of the framework in real-world scenarios.


**Questions:**

1. It would be interesting to see if the proposed model can detect objects beyond the 200 categories in ScanNet-200. Some visualizations or illustrations that demonstrate the upper bound of the proposed method would help readers understand its potential capabilities.

**Limitations:**

The paper doesn't explicitly discuss the limitations of the proposed method.

---

> ### Author Rebuttal · Authors · 2023-08-10
>
> **Q1 Model complexity:**
>
> During testing, our method only adds the CLIP-text encoder and one more distillation head to 3DETR. Therefore, the parameters are just slightly more than 3DETR [19] backbone (6.99M vs. 5.81M), which demonstrates that our designs will not bring many extra computations.
>
> **Q2 Failure cases:**
>
> Thanks for the suggestions. We will clarify the failure cases in the updated version. As shown Fig. C of the rebuttal PDF, failure cases in challenging scenes can be observed for similar semantic categories. For instance, in the second row, a 'coffee table' is misclassified as a 'table'. This issue may be further improved by introducing a stronger classifier than CLIP in future work. We will add the failure case analysis in the revision.
>
> **Q3 How the convergence rate is affected when the novel object list is similar to the ScanNet-200 dataset:**
>
> Following your suggestion, we conducted experiments on SUN-RGBD with the categories of Scan-Net200 as the super-category list for 3D-NOD. As shown in Fig. A of the rebuttal PDF, the training process of our base model, denoted as 'Distillation', is corresponding to the blue curve. The orange curve represents $AP_{Novel}$ of the model trained using 3D-NOD with the LVIS class list, while the green curve indicates the model trained using 3D-NOD with the ScanNet200 class list. From the figure, we can observe that the convergence rate is not significantly affected when replacing the category list with the ScanNet200 list. Notably, the green curve is slightly lower than the orange curve, which is probably because of the significant change in the number of classes, i.e., from a small number of classes (200 categories) to a much larger number of classes (1.2k categories) in the LVIS class list.
>
> **Q4 Comparison with a fully supervised ScanNet-200 method:**
>
> Following your suggestion, we trained a fully supervised model on ScanNet-200. The mAP achieved for all categories is 9.66. Our proposed method achieved mAP of 2.74 across all the categories. As discussed in Sec. 1.2 of the supplementary, the open-vocabulary 3D detection ability of our approach remains limited when evaluated on a larger number of categories (like 200 categories) despite a state-of-the-art performance.
>
> **Q5 Visualization to see if the proposed model can detect objects beyond the 200 categories in ScanNet-20:**
>
> As shown in Fig. B of the rebuttal PDF, our methods can detect the objects belonging to the 'folder' category and beyond the 200 categories in ScanNet-200, which are labeled by blue boxes in the color image of the second row. It shows that our methods have the potential to extend the application to larger vocabularies.
>
> **Q6 Limitations:**
>
> As we discussed in Q4 above, considering the pure point clouds are less discriminative than color images, the open-vocabulary detection ability for larger vocabulary is still limited, which is a common challenge for related 3D open-vocabulary scene understanding [R1, R2, R3].
>
> [R1] Songyou Peng, Kyle Genova, Chiyu Jiang,Andrea Tagliasacchi, Marc Pollefeys, Thomas
> Funkhouser, etal. Openscene: 3d scene understanding with open vocabularies. arXiv preprint
> arXiv:2211.15654,2022.
>
> [R2] Bo Liu,Shuang Deng,Qiulei Dong,and Zhanyi Hu. Language-level semantics conditioned 3d
> point cloud segmentation. arXiv preprint arXiv:2107.00430,2021.
>
> [R3] Runyu Ding, Jihan Yang, Chuhui Xue,Wenqing Zhang,SongBai, and Xiaojuan Qi. Language-driven open-vocabulary 3d scene understanding. arXiv preprint arXiv:2211.16312,2022.

---

> > ### Comment · Reviewer_zhog · 2023-08-18
> > **Thanks for the authors' rebuttal**
> >
> > The rebuttal has addressed my concerns well. I raise the rating to weak accept.

---

> > > ### Author Response · Authors · 2023-08-18
> > >
> > > Dear Reviewer zhog, thank you very much for reviewing our paper, providing valuable suggestions, and carefully reading the rebuttal. We appreciate your time and effort. Your recognition of our work is truly encouraging.

---

### Official Review · Reviewer_Qxx5 · 2023-07-10

**Soundness:** 3 good
**Presentation:** 3 good
**Contribution:** 2 fair
**Rating:** 5
**Confidence:** 5

**Summary:**

This paper proposes an end-to-end open-vocabulary 3D object detection (OV-3DDet) framework which can learn to localize and classify novel objects simultaneously.  To achieve OV-3Det, the author designed a 3D Novel Object Discover (3D-NOD) strategy which utilizes both the 3D box geometry priors and 2D semantic open-vocabulary priors to generate pseudo box labels of the novel objects and develop a Discovery-Driven Cross-Modal Alignment (DCMA) module based on discovered novel boxes, to align feature spaces between 3D point cloud and text modalities for better feature alignment.

**Strengths:**

1. The key difficulty of Open-vocabulary 3D Object detection lies in the localization of novel categories. The paper proposes a novel pseudo novel object annotation generation approach, 3D Novel Object Discovery (3D-NOD) by utilizing the cross-priors from both 3D and 2D domains. Different from previous work, the approach discards the reliance on any additional 2D detection model. The model can learn to localize more objects by updating the novel box label pool.
2. The author provides a detailed analysis of the proposed open-vocabulary detection framework and verifies the benefits of each proposed module. The paper conducts relatively detailed ablation experiments for each proposed module and provides a clear introduction to the implementation details within each module.
3. This paper is relatively clear and easy to follow.

**Weaknesses:**

1. There are minor grammar and spelling issues in the paper, such as:
    1. Line 161:  We then combine both the objections (-> objectness) p_n from the 3D geometry priors a xxxxx.
    2. Line 235: For feature alignment, we randomly selected (-> select) an additional 32 object queries to be involved in the alignment with the CLIP model.

2. Cross-modal feature alignment is common in previous open-vocabulary 3D object detection methods, such as OV-3DET. From my perspective, the DCMA method in the paper demonstrates that not only at the feature level for seen class objects but also on textual alignment for novel class objects should be considered in point-cloud-text feature alignment. From the overall description, it appears that this is not a novel solution but rather an integrated utilization of previous work.
3. The novel box labels generated by the 3D-NOD approach originate from inaccurate 3D bounding boxes produced by the model during training, which are filtered based on certain conditions. These generated novel box labels still have issues in terms of accuracy in both position and size, which limits the final novel object localization accuracy.
4. In Section 3.2, there is inconsistency in the use of symbols during the description of details, and it is rather complex. For example, $l_{n}^{3D}$ and $l_{n}^{2D}$ are unnecessary because people can understand what they are; the image encoder $E_{I,n}^{CLIP}$ can be simplified as $E_{img}$.

**Questions:**

1. In line 186, the author claims that "even if the box covers a background region, the class-agnostic distillation still narrows the distance between the two modalities, leading to a more general alignment on different modalities and scenes." However, I didn't see any experiments to support such claim.
2. In line 235, the paper states "For feature alignment, we randomly selecte an additional 32 object queries to be involved in the alignment with the CLIP model."  Why do we need to select additional 32 object queries to conduct the feature alignment with CLIP model, not using the 128 object queries?
3. Figure 3 shows that the AP_novel keeps stable and AR_novel even decreases after 1080 training epochs. Why are the long epoch training settings used for the final model ?
4. The paper lacks some experiments to demonstrate the upper bound of the detection performance. For example,  the setting of training a class-agnostic 3DETR model on all the categories (base + novel) annotations and using CLIP to  classify the object during inference could be added to demonstrate the how well the cross-modal feature is aligned.
5. Is it possible to show the cross-dataset evaluation performance, for example, training on ScanNet and test on SUN RGBD?

**Limitations:**

The generalization ability of this method is unknown. For example, the open-vocabulary ability remains unknown or limited when more categories (e.g., 200) are introduced.

---

> ### Author Rebuttal · Authors · 2023-08-10
>
> **Q1 Minor grammar issues:**
>
> Thank you very much for your careful review. We will fix the minor issues in the revision.
>
> **Q2 Novelty of DCMA:**
>
> We design the Discovery-Driven Cross-Modal Alignment (DCMA) in conjunction with the 3D-NOD to achieve joint learning of 3D object box discovery and cross-modal class-agnostic alignment. As far as we know, it is the first work that explores novel 3D object box discovery for open-world 3D object detection. The significant difference in terms of class-agnostic alignment with the related work OV-3DET [15] is elaborated in detail in the answer for Q1 of Reviewer DwKh.
>
> **Q3 Limitation from the inaccurate 3D bounding boxes:**
>
> As shown in Tab. 3 of our main paper, compared with the model trained without 3D-NOD (i.e., the '0.0&0.0' baseline), the different thresholds for semantic and geometric priors for 3D-NOD all bring significant improvements (more than 70%) no matter the presence of noisy boxes generated, showing that our method is not sensitive to these filtering conditions. We also agree with the reviewer that if we can further reduce the noisy 3D boxes in the discovery stage, our 3D open-world detection performance can be further boosted since the cross-modal alignment will be more accurate and effective.
>
> **Q4 Inconsistency of the symbols:**
>
> Thank you very much. we will carefully refine this part in the revision.
>
> **Q5 Experiments about the agnostic distillation:**
>
> To evaluate this point, we further conducted experiments to compare the difference between the
> class-agnostic and class-specific distillation, as shown in Tab. D of the rebuttal PDF. As discussed in the main paper, the key difference is whether the method considers the category labels for GT 3D boxes when constructing the distillation objective. In Tab. D of the rebuttal PDF, the $\text{AP}_{Novel}$ of 'Class-agnostic Distillation' outperforms 'Class-specific Distillation', further proving that the class-agnostic distillation offers better generalization for novel categories. We will add the results and discussions in the updated version.
>
> **Q6 Unclear description of the additional 32 object queries:**
>
> We are sorry for the unclear description. To clarify, the total number of object queries remains 128, and we do not increase the number of object queries of the detection backbone. As described in L211-L212 of the main paper, the contrastive alignment is conducted for the object queries that match the box labels from both GT and the discovered novel box pool. Considering that the discovered and GT boxes in a scene are limited, and a larger number of object queries can effectively facilitate contrastive learning, aside from the queries that match GT boxes, we choose an additional 32 object queries from the 128 queries. So 'Additional 32 object queries' are still included in the 128 object queries. Besides, the reason for not performing a contrastive alignment on all 128 queries is to avoid the contrastive learning being dominated by the background boxes, which harms the discovery of novel foreground objects.
>
> **Q7 The reason why the long epoch training settings are used:**
>
> From the observations, we noticed that in the early epochs, the training does not converge well, and the model tends to generate more boxes to cover more foreground objects, thereby increasing AR. However, it will also introduce more noisy boxes that cover the background, while later during the 3D-NOD stage, these noisy boxes cause incorrect discovery of novel object boxes, thus resulting in a worse performance during the 3D-NOD stage. To address this issue, before the 3D-NOD discovery process, we keep training the model for 1080 epochs to reduce the presence of noisy boxes. During these training epochs, both noisy and foreground boxes were reduced, leading to a decrease in AR while maintaining a stable AP. Despite the reduction in foreground boxes, we observed that with fewer noisy boxes, the 3D-NOD algorithm could quickly discover more foreground boxes, as demonstrated in Fig. 3 of our main paper. This led to a significant increase in both AP and AR. Besides, in 3DETR [19], they also train the model for 1080 epochs to make sure the training reaches a good convergence.
>
> **Q8 Upper bound of training a model on all the categories:**
>
> Thank you for the suggestion. We conducted such an experiment to verify the performance upper bound. As shown in Tab. E of the rebuttal PDF, while our method demonstrates significant improvements in 3D novel object detection, compared with the upper bound, there is still room for further investigation.
>
> **Q9 Cross-dataset evaluation:**
>
> Following the suggestion, we have evaluated the cross-dataset performance. The comparisons between the cross-dataset and the same-dataset performances in terms of $AP_{Novel}$ are 3.88 vs. 6.12 on ScanNet and 2.64 vs. 6.71 on SUN RGBD. The lower performances of the cross-dataset evaluation can be due to the data domain gap from the utilization of different RGB-D sensors.
>
>
> **Q10 The generalization ability when more categories (e.g., 200) are introduced:**
>
> To study the generalization ability of our model for more categories, we have conducted experiments shown in Fig. A of the Supplementary. From the results, a gradual decrease in terms of both $AP_{Mean}$ and $AP_{Novel}$ is observed, as the number of test categories increases. When it increases to 200 categories, $AP_{Mean}$ and $AP_{Novel}$ are reduced to 4.68% and 2.85%, respectively. This is a common phenomenon in 3D open-world detection [15].

---

> > ### Author Response · Authors · 2023-08-19
> >
> > Dear Reviewer Qxx5, thank you sincerely for your time and efforts. We greatly appreciate your valuable comments. If you have any questions regarding the rebuttal, we would be delighted to have a discussion.

---

> > > ### Author Response · Authors · 2023-08-21
> > >
> > > Dear Reviewer Qxx5, thank you very much for providing your valuable comments. As the deadline is approaching, we would like to kindly inquire if our discussions have addressed your concerns. If you have any further questions, we would be happy to continue our conversation. If our response has cleared your concerns, we would greatly appreciate it if you could consider updating your score and leaving your feedback. Thank you again.

---

### Official Review · Reviewer_DwKh · 2023-07-10

**Soundness:** 2 fair
**Presentation:** 3 good
**Contribution:** 2 fair
**Rating:** 6
**Confidence:** 4

**Summary:**

This paper is about addressing open-vocabulary 3D object detection, which describes a method aiming for two key challenges, localization and classification for novel objects in point-cloud. The key components are 3D novel-object discover module and the discovery-driven cross-modal alignment module. The first module tries to find as many as possible novel objects based on both geometry and semantic while the second module attempts to align the features of different modalities. The proposed method shows performance improvements compared with the designed baselines.

**Strengths:**

The paper is well-written and shows superior performance improvement compared with the designed baseline.

**Weaknesses:**

I have concerns about both the method novelty and the experiment details. Regarding the method novelty, the key module DCMA is very similar to [15], the difference seems that instead of using contrastive learning, the authors proposed to use distillation loss. However, there is neither insightful discussion nor experiments to analyze why the proposed method is better.

Regarding experiments, [15] is discussed to be similar to this paper in the introduction while no more comparison in experiments, the authors should compare either the whole model or the key components.
On the other hand, I also have concerned about the fairness of the experiments based on 3DETR, as far as I know, 3DETR has much better performance on both SUNRGBD and ScanNet datasets. For the "base" category setting of Table 1, the performance is extremely low, does it mean using CLIP will degenerate the performance of 3DETR? How about directly using the same setting? Furthermore, it is necessary to evaluate the performance on top of different point-cloud detectors instead of only 3DETR.

**Questions:**

My main concern is about the discussion to [15] and more necessary experimental analysis. Please refer the weakness part.

**Limitations:**

The work is only designed based on 3DETR, an indoor point-cloud detector. It should be evaluated based on more detectors.

---

> ### Author Rebuttal · Authors · 2023-08-09
>
> **Q1 Novelty of DCMA:**
>
> Our proposed Discovery-Driven Cross-Modal Alignment(DCMA) has two key differences from the contrastive learning in [15]:
>
> (i) Joint Learning with 3D Novel Object Discovery (3D-NOD): DCMA is simultaneously learned with our 3D-NOD, enabling generalization to a broader range of novel categories during the discovery process, which can be demonstrated by comparing '3D-NOD + DCMA (full)' over '3D-NOD + Distillation' and '3D-NOD + Distillation \& PlainA' shown in Tab. 1 of the main paper. In contrast, [15] performs alignment for additional categories by utilizing an additional large pretrained 2D open-vocabulary detection model [34] to generate 3D pseudo labels, making a substantial difference in learning and obtaining open-world knowledge. To the best of our knowledge, the collaborative learning of classification and localization for novel objects clearly highlights a novel contribution in open-vocabulary 3D detection (OV-3DDet).
>
> (ii)	Class-Agnostic Distillation during Discovery: [15] proposes a class-specific contrastive learning strategy, while we present a class-agnostic distillation that can more effectively align cross-modal features during the discovery process. The class-specific contrastive learning encourages features belonging to the same categories to be as similar as possible. However, during the discovery process, the discovered boxes may cover background regions. Since CLIP cannot classify the background, boxes on background with higher scores may be assigned to incorrect categories, thus leading to severe feature misalignment by class-specific contrastive learning. Conversely, our class-agnostic distillation aims at making the 3D object query features align with their corresponding projected 2D image features. The distillation can also cover 3D object boxes on the background. So the distillation can be effectively performed for the foreground and background simultaneously. To show our advantage compared to class-specific contrastive learning, we replace class-agnostic distillation with class-specific contrastive learning in our methods. As shown in Tab. A of the rebuttal PDF, the contrastive loss encourages more boxes to be classified as the foreground, resulting in a larger recall. However, as discussed, it also encourages more background boxes to be misclassified as foreground objects, leading to a lower overall mAP compared to our class-agnostic distillation.
>
> **Q2 Comparison with [15]:**
>
> We would like to highlight that the setting in our methods is significantly different from [15]. In our setting, we train the model using annotations from a small number of base categories (10 categories) and learn to discover novel categories during training. We evaluate our model in a large number of novel categories (50 in ScanNet, and 46 in SUN-RGBD). However, [15] relies on an large-scale pretrained 2D open vocabulary detection model, i.e., OV-2DDet [34] model, to generate pseudo labels for novel categories, and evaluate the model on only 20 categories. Thus, because of the setting differences, and the prior model utilized, a direct comparison with [15] is not straightforward. To provide a comparison with [15] as requested, thanks to the releasing the codes from the authors of [15] to generate pseudo labels on ScanNet, we can directly retrain our method following the same setting of [15] on ScanNet, and provide a fair comparison with [15] on Tab. B of the rebuttal PDF. As can be observed, our method achieves clearly better mean AP (1.3 points improvement over all the categories), validating the superiority of our method.
>
> **Q3 Comparison with 3DETR [19]:**
>
> We want to further clarify that, the settings on base categories between ours and 3DETR are different. Our method targets 3D open-vocabulary (OV) detection, and thus we follow a typical setting for base categories in 2D-OV detection frameworks, i.e., using top-10 categories with the highest number of training samples as the base categories. Because of the different settings, direct comparison on mAP of 10 categories (including 5 different categories) is not fair. To compare our method and 3DETR fairly, we can compare the mAP of the 5 shared categories. As shown in Tab. C of the rebuttal PDF, it can be clearly observed that our method achieves better performance than 3DETR in terms of the mean AP of the shared categories (58.51 vs. 57.36), which may confirm the benefits of using CLIP features for learning more discriminative features. Besides, according to the reviewer’s suggestion, we also conducted experiments to show the performance comparison between ours and 3DETR considering the same setting of the base categories. Limited by the computation resources, the model training is not finished yet. We will try to provide the comparison in the later discussion period once it is finished.
>
> **Q4 Different 3D detectors:**
>
> Thanks for the suggestion. We considered 3DETR because it is a widely employed transformer-based 3D object detection backbone, which is also utilized by related work [15]. Our proposed method contributes a new technique for collaborative learning novel 3D object box discovery and cross-modal alignment, which is independent of different detection frameworks. Because of the time limit of the rebuttal, we need more time to run results using other 3D detection frameworks. We will do our best to provide new results regarding this point in the later discussion period.

---

> ### Author Response · Authors · 2023-08-15
>
> Dear Reviewer DwKh, thank you very much for your valuable suggestions and time. Regarding the experiment mentioned in Q3, the model training is finished, and our method achieves clearly better performance than 3DETR using the same setting, confirming the benefits of using CLIP. May we ask if you would like to see the detailed results? According to the discussion rule, we can provide the results when you are interested.

---

> > ### Comment · Reviewer_DwKh · 2023-08-16
> >
> > Dear Authors
> >
> > Yeah, please provide the results. A clarification also should be addressed as that why 3DETR performs not as well as the original paper reported, for example, 3DETR achieves 62&58 AP25 on ScanNet&SunRGBD, while your "base" setting only got ~30. Why the model performs much worse. So I have concerns that CLIP degenerates performance. What I expect is that your method will not degenerate the "BASE" setting while is able to improve the detection for novel classes.
> >
> > Besides, different class-splitting settings should also be conducted, e.g.,  Do the different "base" classes and "novel" classes have different improvements? Could you also provide it?
> >
> > Thanks!

---

> > > ### Author Response · Authors · 2023-08-19
> > >
> > > Dear Reviewer DwKh, thank you for your reply and valuable suggestions. We have conducted experiments these days for your questions and are now providing you with further answers:
> > >
> > > **Q1 Regarding the results of our model trained in the close-vocabulary 3DETR settings:**
> > >
> > > As suggested, we trained our method using the same 10 categories as 3DETR, without considering the novel categories (*i.e.*, keeping the cross-modal alignments and disabling 3D-NOD for novel box discovery), and report the results in Tab. A shown below. It can be observed that our method achieves an $\text{AP}_{0.25}$ of 57.98, which outperforms 3DETR, suggesting the potential benefits of incorporating CLIP.
> > > |         |                |       |       |       |       |       |        |         |            |           |         |
> > > |:-------:|:-------:|:-------:|:-------:|:-------:|:-------:|:-------:|:-------:|:-------:|:-------:|:-------:|:-------:|
> > > | Methods | **Mean**  | chair | table | desk  | bed   | sofa  | toilet | dresser | nightstand | bookshelf | bathtub |
> > > | 3DETR   | 56.80          | 68.00 | 50.00 | 28.70 | 81.80 | 58.30 | 90.30  | 28.60   | 56.60      | 27.50     | 77.60   |
> > > | Ours    | **57.98** | 69.33 | 49.01 | 28.79 | 85.54 | 62.88 | 91.61  | 29.65   | 61.89      | 25.88     | 75.27   |
> > >
> > > &emsp;Tab. A: Performance comparison with the 3DETR [19] in the same close-vocabulary 3DETR settings.
> > >
> > > Regarding the mentioned $AP_{0.25}$ performance of 3DETR, *i.e.*, 58 on SUNRGBD, we have identified an inconsistency between the camera-ready version [A] and the arXiv version [B] of the 3DETR paper. Specifically, Tab. 1 in the camera-ready version [A] reports an  $AP_{0.25}$ of 56.8 for 3DETR, while Tab. 1 in the arXiv version reports an $AP_{0.25}$ of 58.0. To further investigate the inconsistency, (i) we carefully check the detailed comparisons in Tab. 11 in the supplemental material of both **the camera-ready and the arXiv versions**, which reveal that the  $AP_{0.25}$ of 3DETR is consistently reported as 56.8 in both versions. (ii) we also carefully check the per-class AP comparisons in Tab. 9 in the supplemental material of both the camera-ready and the arXiv versions. Then we calculate the mean AP of the 10 categories, resulting in a mean AP of 56.74. Based on all of the above details, we consider  $AP_{0.25}$ of 56.8 shown in the camera-ready version as the performance of 3DETR, and use this number for our comparison, as shown in Tab. A above.
> > >
> > > [A] Ishan Misra, Rohit Girdhar, and Armand Joulin. An end-to-end transformer model for 3d object detection. In Proceedings of the IEEE/CVF International Conference on Computer Vision, pp. 2906–2917, 2021.
> > > [B] Ishan Misra, Rohit Girdhar, and Armand Joulin. An end-to-end transformer model for 3d object detection. arXiv preprint arXiv:2109.08141, 2021
> > >
> > > **Q2 The setting difference between the $AP_{Base}$ of '3D-CLIP' and the AP in the 3DETR paper:**
> > >
> > > The main reason why there is a gap between the $AP_{Base}$ of '3D-CLIP' and the AP in 3DETR paper is that **the category setting is significantly different**. As we clarified for this point in the answer for Q3 in the rebuttal, following the typical category-splitting strategy widely considered in 2D open-vocabulary object detection, we use top-10 categories with the highest number of training samples as the base categories. We show the category names in Tab. C in the rebuttal PDF. There are **5 categories in our base category setting that do not overlap with the 3DETR category setting**. These 5 categories may be harder for point cloud object detection. For instance, the $AP_{0.25}$ of the category 'box' and 'computer' is only 0.57 and 2.41, respectively. This is probably because their geometric structures mostly resemble simple cubes and lack distinctive information. The worse performance in these hard categories leads to a lower mean AP in our setting. Following your suggestions, we also conducted the experiments with the same 10 categories of 3DETR as our base categories shown in the answer to the next question.
> > >
> > > Due to the space limitation, the answers to the next questions are provided in **the following reply**.

---

> > > > ### Author Response · Authors · 2023-08-19
> > > >
> > > > **Q3 Will CLIP degrade the performances?**
> > > >
> > > >  We answer this question from the following three aspects:
> > > >
> > > > (i) When we **do not consider novel categories** and train our model in the close-vocabulary setting (*i.e.*, keeping the cross-modal alignments and discarding the 3D-NOD), CLIP does not degrade the performance as shown in Tab. A in last reply. This is because, during the training, the CLIP knowledge can be adapted for 3D detection on the base categories by learning the cross-modal alignment.
> > > >
> > > > (ii) When we **consider novel categories and directly apply pre-trained CLIP in the testing stage** to achieve open-vocabulary detection (*i.e.*, '3D-CLIP', the open-vocabulary baseline model), the mAP of the base categories of '3D-CLIP' is lower than the AP of 3DETR, as shown in Tab. B below. The base categories of our method in this experiment are the same as the categories in the 3DETR paper. It is reasonable because the pre-trained CLIP is directly used for open-vocabulary classification in the testing. There is no cross-modal alignment between CLIP and the target data distribution during training. However, '3D-CLIP' can detect objects of novel categories and obtain an $AP_{Novel}$ of 2.39, while the 3DETR cannot perform novel object detection, further confirming the benefits of using CLIP for the problem.
> > > >
> > > > (iii) When we **consider novel categories and involve CLIP during training in our designed method**, the $AP_{Novel}$ and $AP_{Base}$ of our method are both significantly higher than the open-vocabulary baseline model '3D-CLIP', as shown in Tab. B below. While $AP_{Base}$ of our method is slightly lower than 3DETR, we think it is a reasonable phenomenon since 3D-NOD may also possibly discover noisy 3D boxes and the method needs to jointly learn both the base and the novel categories which is more challenging compared to 3DETR that only learns the base categories.
> > > >
> > > > |Methods | $\text{AP}_{Novel}$ | $\text{AP}_{Base}$ |
> > > > |:-------:|:-------------------:|:------------------:|
> > > > | 3DETR   | -                   | 56.80              |
> > > > | 3D-CLIP | 2.39                | 41.64              |
> > > > | Ours    | **5.13**                | **55.26**              |
> > > >
> > > > ***Tab. B:** Performance comparison using the same 10 categories in the 3DETR paper as the base categories. '3DETR' results show the performance of the original camera-ready version of 3DETR. '-' indicates that the close-vocabulary 3DETR cannot perform detection on novel categories. '3D-CLIP' is our open-vocabulary baseline model. Our method achieves significant improvements on both $AP_{Novel}$ and $AP_{Base}$ upon the baseline.*
> > > > &nbsp;
> > > >
> > > > **Q4 Other class-splitting settings:**
> > > >
> > > > We follow the suggestion of the reviewer and set the same 10 categories of 3DETR as our base categories. As shown in Tab. B above, our method also achieves significant improvements on both $AP_{Novel}$ and $AP_{Base}$ compared to the baseline model, further demonstrating the effectiveness of our proposed method.

---

> > > > > ### Author Response · Authors · 2023-08-21
> > > > >
> > > > > Dear Reviewer DwKh, thank you very much for providing your valuable comments. As the deadline is approaching, we would like to kindly inquire if our discussions have addressed your concerns. If you have any further questions, we would be happy to continue our conversation. If our response has cleared your concerns, we would greatly appreciate it if you could consider updating your score and leaving your feedback. Thank you again.

---

### Author Rebuttal · Authors · 2023-08-10

We thank all the reviewers for their detailly suggestions and encouraging comments: **"The idea makes sense"** and **"is well-motivated"** (Reviewer zhog, Reviewer goao). **"Discard the reliance on any additional 2D detection model"** (Reviewer Qxx5, Reviewer zhog). Bring **"superior performance improvements"** (Reviewer DwKh, Reviewer zhog). The paper is **"well-written"** and **"easy to follow"** (Reviewer DwKh, Reviewer Qxx5).

To make the discussion clearer, we will answer all the questions one by one for each reviewer, respectively. Note that all the tables and figures we prepared for this rebuttal are in **the PDF for the rebuttal**. Please have a check.



In this global response, we claim the potential negative societal impact, which will be updated in the new version.

**Potential negative societal impact:**


In our work, we train our methods only on public datasets [4, 24], ensuring that no private data is utilized. After releasing the codes and pre-trained models, some people may possibly consider using our method for large-scale open-world training involving personal data, causing the data privacy issue. We will explicitly cover this issue when we release the project and restrict any utilization of our project that may cause a potential negative societal impact.

---

### Author Response · Authors · 2023-08-15

Dear reviewers, we sincerely appreciate your time and valuable comments. If you have any questions regarding the rebuttal, we would be delighted to discuss them with you.

---

### Decision · Program_Chairs · 2023-09-21

**Decision:**

Accept (poster)

**Comment:**

The paper initially receives mixed ratings (two positive and two negative reviews). After the rebuttal and discussion phases, two negative reviews are upgraded to borderline acceptance. The AC took a close look at the paper, reviews, and rebuttals.

The tackled problem is of great interest and the authors have demonstrated a technically sound framework, especially the proposed way to discover and localize novel 3D objects via alignment between 2D and 3D spaces. Results are convincing and promising with extensive experiments. Overall, the AC finds that all the concerns are addressed well in the rebuttal and response, especially for questions from reviewer DwKh and goao who provided negative ratings initially. Therefore, the AC recommends the "acceptance" rating.